

# HyPhAI v1.0: Hybrid Physics-AI architecture for cloud cover nowcasting

Rachid El Montassir[1], Olivier Pannekoucke[1,2,3], and Corentin Lapeyre[1]

[1]CERFACS, Toulouse, France
[2]INPT-ENM, Toulouse, France
[3]CNRM, Université de Toulouse, Météo-France, CNRS, Toulouse, France

**Correspondence:** Rachid El Montassir (elmontassir@cerfacs.fr), Olivier Pannekoucke (olivier.pannekoucke@meteo.fr)

**Abstract.** This work proposes a hybrid approach that combines Physics and Artificial Intelligence (AI) for cloud cover nowcasting. It addresses the limitations of traditional deep learning methods in producing realistic and physically consistent results that can generalize to unseen data. The proposed approach enforces a physical behaviour. In the first model, denoted HyPhAI-1, a multi-level advection dynamics is considered as a hard constraint for a trained U-Net model. Our experiments show that the hybrid formulation outperforms not only traditional deep learning methods, but also the EUMETSAT Extrapolated Imagery model (EXIM) in terms of both qualitative and quantitative results. In particular, we illustrate that the hybrid model preserves more details and achieves higher scores based on similarity metrics in comparison to the U-Net. Remarkably, these improvements are achieved while using only one-third of the data required by the other models. Another model, denoted HyPhAI-2, adds a source term to the advection equation, it impaired the visual rendering but displayed the best performance in terms of Accuracy. These results suggest that the proposed hybrid Physics-AI architecture provides a promising solution to overcome the limitations of classical AI methods, and contributes to open up new possibilities for combining physical knowledge with deep learning models.

## 1 Introduction

Meteorological services are responsible for providing accurate and timely weather forecasts and warnings to ensure public safety and mitigate damage to property caused by severe weather events. Traditionally, these forecasts have been based on numerical weather prediction (NWP) models, which provide predictions of atmospheric variables such as temperature, humidity, and wind speed. However, NWP models have inherent limitations in their ability to capture small-scale weather phenomena such as thunderstorms, tornadoes, and localised heavy rainfall events.

To address this limitation, the concept of *nowcasting* has emerged as a valuable tool in meteorology (Lin et al., 2005; Sun et al., 2014). Nowcasting refers to the process of using recently acquired high-resolution observations to generate short-term forecasts of weather conditions, typically on a timescale of minutes to a few hours. Nowcasting techniques exploit various observational data sources, including radar, satellite, lightning, and ground-based observations, to generate real-time estimates of weather conditions and can take advantage of these recent data to significantly outperform NWP on short lead times (Lin et al., 2005; Sun et al., 2014).





Cloud cover nowcasting is a critical component of weather forecasting. It is used to predict the likelihood of precipitation, thunderstorms, and other hazardous weather events. Accurate cloud cover forecasts on a short timescale are particularly important for weather-sensitive applications such as aviation, agriculture, and renewable energy production.

Traditionally, cloud cover forecasting has been done using physics-based methods, relying on the laws of physics to model the evolution of cloud cover, e.g. cloud motion vectors as in Bechini and Chandrasekar (2017); García-Pereda et al. (2019),
optical flow (Wood-Bradley et al., 2012), or NWP-based Data Assimilation (Ballard et al., 2016). However, with the recent advances in Artificial Intelligence (AI) and Machine Learning (ML), data-driven methods have become increasingly popular for this type of tasks (e.g. Espeholt et al., 2022; Ravuri et al., 2021; Trebing et al., 2021; Ayzel et al., 2020; Berthomier et al., 2020; SHI et al., 2015).

Among these data-driven methods, Long Short-Term Memory (LSTM) networks, introduced by Hochreiter and Schmidhu-
ber (1997), stand out. LSTMs are a type of recurrent neural network capable of learning long-term dependencies, they are useful for time series predictions as they can learn from past entries to predict future values. In tasks involving multidimensional data, they are commonly used with convolutional layers, forming what is known as a convolutional LSTM. This network excels in capturing spatio-temporal correlations compared to fully-connected LSTMs (SHI et al., 2015). Spatio-temporal LSTM (Wang et al., 2018) increases the number of memory connections within the network, allowing an efficient flow of spatial informa-
tion. This model was further optimised by adding stacked memory modules (Wang et al., 2019). U-Net is another popular architecture, it was initially designed by Ronneberger et al. (2015) for biomedical image segmentation. Unlike LSTMs, U-Net has no explicit memory modelling, yet it has shown good performance for a binary cloud cover nowcasting task as shown in Berthomier et al. (2020). Additionally, it has found application in precipitation nowcasting as highlighted by Ayzel et al. (2020), and a modified version was used for a similar task in Trebing et al. (2021).

Machine learning models hold great promise for addressing scientific challenges associated with processes that cannot be fully simulated, either due to lack of resources or complexity of the physical process. However, their application in scientific domains had faced challenges, including constraints related to large data needs, difficulty in generating physically coherent outcomes, limited generalisability, and issues related to explainability (Karpatne et al., 2017). To overcome these challenges, incorporating physics into ML models is of paramount importance. It leverages the inherent structure and principles of physical
laws to enhance model interpretability, handle limited labelled data, ensure consistency with known scientific principles during optimisation, and ultimately improve the overall performance and applicability of the models, making them more likely to be generalisable to *out-of-sample* scenarios. As discussed by Willard et al. (2022), the hybridisation available techniques leverage different aspects of ML models, e.g. the cost function, the design of the architecture or the weights' initialisation.

A common method to ensure the consistency of ML models with physical laws is to embed physical constraints within the
model's loss function (Karpatne et al., 2017). This involves incorporating a physics-based term weighted by a hyperparameter, alongside the supervised error term. This addition enhances prediction accuracy and accommodates unlabelled data. It has proven to be effective in addressing a range of problems, including uncertainty quantification, parameterisation, and inverse problems (Daw et al., 2021; Jia et al., 2019; Raissi et al., 2019). However, one drawback lies in the challenge of appropriately tuning the hyperparameter.



Given the necessity for an initial choice of model parameters in many ML models, researchers explore ways to inform the initial state of a model with physical insights. One possible way is transfer learning, where a pre-trained model is fine-tuned with limited data (Jia et al., 2021). Additionally, simulated data from physics-based models can be employed for pre-training, akin to methods used in computer vision. This technique has found application in diverse fields, including biophysics (Sultan et al., 2018), temperature modelling (Jia et al., 2019), and autonomous vehicle training (Shah et al., 2017). However, this

method requires the assumption that the underlying physics of the simulated data aligns with the real-world data.

To address imperfections in physics-based models, a common strategy is residual modelling. Here, an ML model learns to predict the errors (residuals) made by the physics-based model (Forssell and Lindskog, 1997). This approach leverages learned biases to correct predictions. However, it does not have the ability to enforce physics-based constraints, as it primarily deals with errors rather than physical states.

An advanced variation of residual modelling involves the integration of physics-based models and ML models. In scenarios where the dynamics of Physics are fully defined, a straightforward method involves using the output of a physics-based model as an input to an ML model. This approach has demonstrated enhanced predictions in tasks such as lake temperature modelling (Daw et al., 2021). However, in cases where a physical model contains unknown elements requiring coupling with an ML model for joint resolution, a viable strategy involves substituting a segment of a comprehensive physics-based model

with a neural network. An illustrative example is found in sea surface temperature prediction, where de Bezenac et al. (2018) employed a neural network to estimate the motion field. In alignment with this strategy, our study proposes leveraging physical knowledge based on the advection equation to address the cloud cover nowcasting task. This results in simulating the advection of clouds by winds while using a neural network to estimate unknown variables, such as the two components of the velocity field.

Moreover, our study introduces an additional requirement - cloud type classification. Specifically, our dataset contains cloud cover observations with pre-existing categorical classifications based on cloud types (e.g. very low clouds, low clouds). This necessitates adopting a probabilistic approach in our hybrid architecture, which, to the best of our knowledge, has not been explored in geophysics. Indeed, adopting a probabilistic approach with probability maps allows us to account for the inherent variability and uncertainties associated with the model's predictions. This also provides a more natural framework for such a

classification problem for further extensions of the modelling beyond the advection.

Rather than using the theoretical solution of the equation as proposed in de Bezenac et al. (2018), our hybrid approach solves a system of Partial Differential Equations (PDEs) within a neural network, which make the architecture more flexible. However, it poses some implementation challenges, as explained in Sect. 2. Moving forward, we introduce the hybrid architecture in Sect. 3. Section 4 is dedicated to presenting results and performance analysis compared to state-of-the-art models. Finally, in

Sect. 5, we draw conclusions based on our findings.





## 2 Bridging neural networks and numerical modelling

In this section, we present fundamental components for implementing the proposed hybrid architecture. In Sect. 2.1 we explore the integration of physics within a neural network. We then explain the trainability challenges associated with this architecture in Sect. 2.2. Following that, in Sect. 2.3, we provide a brief introduction to numerical methods for solving PDEs. Finally, in the Sect. 2.4 and Sect. 2.5, we present the method used to approximate derivatives and perform time integration within a neural network.

### 2.1 Combining neural networks and Physics

An artificial neural network is a function $f_\theta$ parameterised by a set of parameters $\theta$. It results from the composition of a sequence of elementary non-linear parameterised functions called layers, that process and transform input data $x$ into output predictions $y$ as follows:

$$y = f_\theta(x). \tag{1}$$

Physics-based models aim to represent the underlying physical processes, or equations, that govern the behaviour of a system. To incorporate physics into the neural network, one possible approach involves feeding the output of the physics-based model as an input to the neural network as follows:

$$y = f_\theta(x, \phi(x_{\text{Phy}})), \tag{2}$$

where $x_{\text{Phy}}$ are the inputs of the physics-based model $\phi$. This method could be effective when the physics-based model is self-contained, and its components are explicitly known. However, it becomes impractical in scenarios where the physics-based model presents unknown variables that need to be estimated. This is the case in the application considered in this work, where the cloud motion field is unknown. In such instances, a more suitable approach is to pursue a joint resolution. Here, the physical model takes the outputs of the neural network and computes the predictions, resulting in a composition of $f_\theta$ and $\phi$ as follows:

$$y = \phi \circ f_\theta(x, x_{\text{Phy}}). \tag{3}$$

In this approach, $\phi$ implicitly applies a hard constraint on these outputs, this might contribute to accelerate the convergence of the neural network during the training process.

Unlike the first method (Eq. (2)), where the physics-based model $\phi$ is passive and not involved in the training procedure, the second method raises some challenges concerning the trainability of the architecture.

### 2.2 Training a neural network

Neural networks learn to minimise a loss function $\mathcal{L}_\theta$ by adjusting its set of parameters $\theta$ using data. The loss function measures the error between the predicted outcomes $\hat{y}$ and the ground truth $y$. It is expressed as

$$\mathcal{L}_\theta = \frac{1}{N} \sum_{k=1}^{N} l(y_k, \phi \circ f_\theta(x_k)), \tag{4}$$





where $N$ is the sample size and $l$ is a measure of the discrepancy between the ground truth $y_i$ and the model's production

associated with the input $x_i$, i.e., $f_\theta \circ \phi(x_i)$. For instance, using $l(a,b) = \|a - b\|^2$ is the measure used to calibrate a regression

model and $\mathcal{L}_\theta$ is then the so-called mean squared error (MSE). The choice of $l$ depends, among other things, on the statistical

model $f_\theta$.

During this training process, an algorithm called backpropagation is used to optimise model parameters. Backpropagation

involves computing the gradient of the loss function with respect to the network's parameters. It indicates how much each

weight contributed to the error. This gradient is then used to update the parameters in the direction that minimises $\mathcal{L}_\theta$, following

a sequential optimisation algorithm such as gradient descent, as described below:

$$\theta_{\text{updated}} = \theta_{\text{old}} - \gamma \nabla \mathcal{L}_{\theta_{\text{old}}}, \tag{5}$$

where $\gamma$ is the magnitude of the descent. In order to perform the backpropagation, we assume that the gradient of the loss

function with respect to the model's parameters could be calculated using the chain rule. This assumption is called differentia-

bility. Indeed, neural networks rely on activation functions and operations that are differentiable, allowing the gradients to be

propagated backward through the network layers.

In this proposed hybrid approach, PDEs are solved to produce model predictions. If the PDE solver includes non-differentiable

steps, the chain rule breaks down, making it impossible to compute gradients within the standard deep learning frameworks.

In what follows, we describe our strategy for modelling and solving PDEs using basic differentiable operations commonly

employed in neural networks.

### 2.3   Numerical methods for partial differential equations

Numerical weather prediction involves addressing equations of the form

$$\partial_t f = \mathscr{F}\left(f, \partial_x f, \partial_x^2 f, \ldots\right), \tag{6}$$

governing the evolution of a univariate or multivariate field $f$ over time. Computers cannot directly solve symbolic PDEs, and

a common approach involves a two-stage process to transform the PDE into a mathematical formulation more suitable for

computational handling. This process begins by discretising the partial derivatives with respect to spatial coordinates, resulting

in an ordinary differential equation. Subsequently, a temporal integration describes the evolution of the system over time.

Spatial discretisation can be performed using several methods, e.g. finite volumes, finite elements, or spectral methods.

However, the simplest one, the finite-difference method, consists in replacing spatial derivatives of $f$ by quantities which

depends on the values of $f$ on a grid. For example, on a 1D periodic domain $[0, L]$ of coordinate $x$, discretised in $N$ grid points

$(x_i)_{[0, n-1]}$ $(x_n = x_0)$, the central difference method of the first-order spatial derivative reads

$$\partial_x f(t, x_i) \approx \frac{f(t, x_{i+1}) - f(t, x_{i-1})}{2\delta x}, \tag{7}$$



where $\delta x = x_{i+1} - x_i$ represents the grid resolution. Following spatial discretisation, Eq. (6) can be written as an ordinary
differential equation as follows:

$$\frac{d\mathbf{f}}{dt} = \hat{F}(\mathbf{f}), \tag{8}$$

where $\mathbf{f}(t)$ is the discretised form of $f$ over the spatial domain, e.g. the vector of grid-point values of $f$ at time $t$, i.e. $\mathbf{f}(t) = (f(t, x_i))_i$ in the 1D domain mentioned above.

For the time integration, various methods can also be employed, e.g. Euler's method or a fourth order Runge–Kutta method
(RK4) (Runge, 1895; Kutta, 1901). These methods differ in their accuracy, stability, and computational cost. An explicit Euler time integration of Eq. (8) reads

$$\mathbf{f}_{q+1} = \mathbf{f}_q + \delta t \hat{F}(\mathbf{f}_q), \tag{9}$$

where $\mathbf{f}_q = \mathbf{f}(t_q)$, with $t_q = q\delta t$ the discretised time of time step $\delta t$.

For the sake of illustration, we consider the advection over the above-mentioned 1D periodic domain, given by the following
equation:

$$\partial_t f + u \partial_x f = 0, \tag{10}$$

where $u$ is a velocity field whose values on the grid are denotes as $(u_i)_{i \in [0, n-1]}$. Applying central difference and an Euler scheme discretisation yields the sequential evolution:

$$f_{q+1,i} = f_{q,i} - \frac{\delta t}{2\delta x} u_i \left( f_{i+1} - f_{i-1} \right). \tag{11}$$

This example illustrates the integration of the advection equation over time using a simple explicit method. However, depending on the problem characteristics and requirements, other time integration schemes may be more suitable.

In this study, we propose to model and solve PDEs within a neural network, e.g. equations of the form Eq. (6). This is achieved by describing the equivalent of spatial and temporal discretisation in the frame of neural network layers, i.e. how it can be implemented in a deep learning (DL) framework as TensorFlow (Abadi et al., 2016) or PyTorch (Paszke et al., 2019).

## 2.4 Finite-difference methods and convolutional layers

To implement a finite-difference discretisation, one viable approach is to employ the convolution operation. For instance, the 1D convolution associated with Eq. (7) can be mathematically written as:

$$(K^1 * f)[i] = \sum_{m=0}^{M-1} K^1[i] f[m+i], \tag{12}$$

where $K^1$ is the kernel or filter used for the convolution and expressed as

$$K^1 = \begin{bmatrix} \frac{-1}{2\delta x} & 0 & \frac{1}{2\delta x} \end{bmatrix},$$





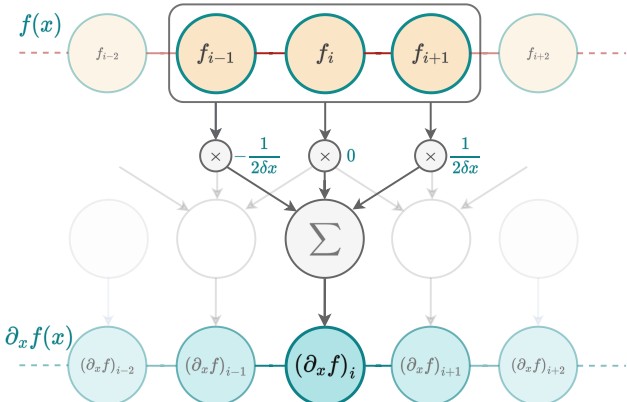

**Figure 1.** In order to calculate the numerical derivative of $f$, a kernel $K^1$ is used to slide across an input vector, which is a discretised version of $f$ with $N$ elements, element-wise multiplying values within its window and summing the results to approximate the derivative at each position. The result is a new vector of size $N-2$ containing the numerical derivative of $f$ (padding at the bounds with zeros or duplicated values in the input vector can be applied to produce an output vector of size $N$). This is equivalent to a convolution between $K^1$ and $f$, and can be reproduced using a 1D convolutional layer with $K^1$ as a kernel.

and $f$ represents the input data. The variable $M$ corresponds to the size of the kernel. It is the number of finite-difference coefficients, also called stencil size. In this case, a three-point stencil is considered ($M=3$). Finally, $*$ is the convolution operator.

This leads to an interesting interaction with DL frameworks. Indeed, convolutional neural networks (CNNs) rely on the operation

$$\text{ConvLayer}(f)\,[i] = \sigma\left(\sum_{m=0}^{M-1} K\,[m]\,f\,[m+i]+b\right),$$

where $\sigma$ is called *activation function* and $b$ is a parameter representing the *bias*. Observing that using $\sigma = \textit{identity}$ and $\text{b}=0$ leads to the same operation as in Eq. (12), one can leverage deep learning frameworks to approximate derivatives, which enables derivative-based operations in neural networks, as shown in Fig. 1. The same principle applies to higher derivative

orders. For any positive integer $\alpha$, we can write the approximation of the $\alpha$-th derivative of $f$ as

$$\partial^\alpha f \approx K^\alpha * f, \tag{13}$$

where $K^\alpha$ are the finite difference coefficients for the $\alpha$-th derivative.

Finally, using convolutions is a straightforward method to model the spatial term of a PDE, also called the trend, as follows

$$\hat{F}(f) = \mathcal{N}(f). \tag{14}$$

This results in a neural network that can be used for time integration.



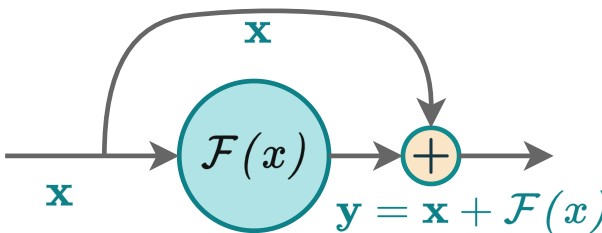

**Figure 2.** Illustration of a residual block

## 2.5 Temporal schemes and residual networks

The time integration expressed in the Eq. (9) can be written by using the neural network implementation $\mathcal{N}$ of the trend as

$$f_{n+1} = f_n + \Delta t \mathcal{N}(f_n). \tag{15}$$

Interestingly, this formulation is very similar to that of a building block commonly used in deep neural networks called a
*residual block* (see Fig. 2), proposed in the ResNet architecture (He et al., 2016). It is formulated as follows:

$$y = x + \mathcal{F}(x), \tag{16}$$

where $x$ is the input to the block, $y$ is the output, and $\mathcal{F}$ is called a *residual function*, made up of multiple neural layers. These
layers represent the difference between the input and output. This function aims to capture the additional information or adjust-
ments needed to transform the input into the desired output. This similarity between residual blocks and time schemes, also ob-
served in Ruthotto and Haber (2020); Chen et al. (2019); Fablet et al. (2017), suggests that the time integration step can be done
inside a neural network, all we need is the residual function, which can be modelled using convolutional layers as shown pre-
viously. Pannekoucke and Fablet (2020) proposed a general framework (called https://github.com/opannekoucke/pdenetgen),
to model a PDE in a neural network form using this method. Residual blocks were originally designed to address vanishing
gradient issues in image classification tasks. Intriguingly, these blocks proved to function similarly to time schemes, where they
introduce small changes over incremental time steps. This challenges the traditional black box perception of neural networks,
although full interpretability is remaining a distant goal.

   We have shown that spatial derivatives of PDEs can be approximated within a neural network in a differentiable way. This
allows us to compute gradients and back-propagate them during the training process. This fundamental knowledge serves as a
foundation for our investigation of novel hybrid Physics-AI architectures. With these established principles, we present in the
following section the proposed hybrid architecture, which is used for cloud cover nowcasting.



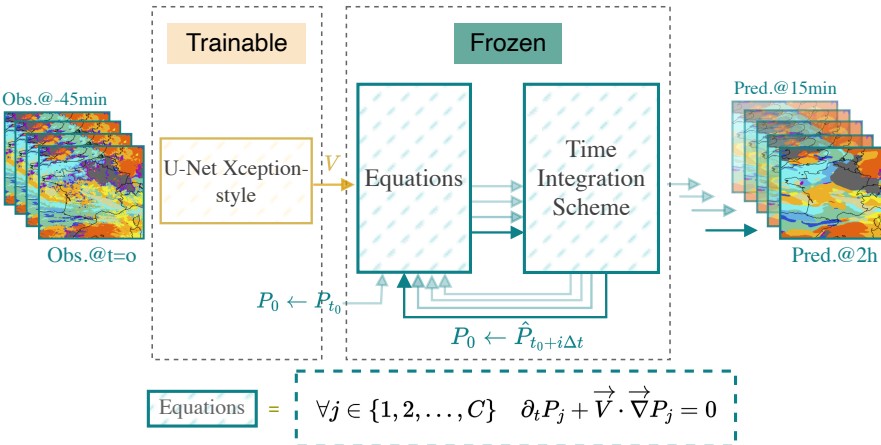

**Figure 3.** HYPHAI-1: The proposed hybrid model consists of a U-Net Xception-style model to estimate the velocity field from the last observations, the estimated velocity field is smoothed using a Gaussian filter. The advection equation is numerically integrated using the 4-th order Runge–Kutta method over multiple time steps. The initial condition ($f_0$) is updated, after each time step, to the current state, allowing the computation of the next state.

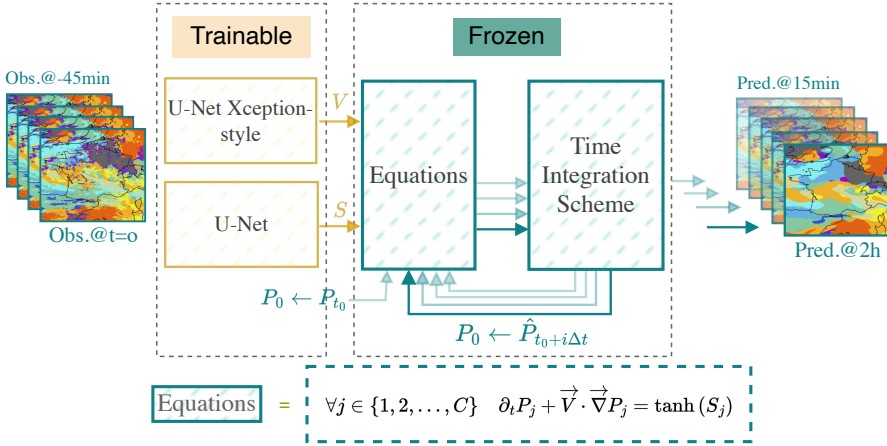

**Figure 4.** HYPHAI-2: The second version of the proposed hybrid model. It consists of a U-Net Xception-style to estimate the velocity field and a second U-Net to estimate the source term from the last observations.

## 3 Proposed HYPHAI model

In this section, we introduce our hybrid architecture, denoted as HYPHAI (an abbreviation for Hybrid Physics-AI), detailed in Sect. 3.1. Section 3.2 explains the different physical modelling approaches investigated in this study. Following that, Sect. 3.3, Sect. 3.4 and Sect. 3.5 sequentially present the training procedure, evaluation metrics, and benchmarking procedure.





## 3.1 The HYPHAI architecture

The proposed HYPHAI architecture, is a dual-component system (see Fig. 3). The first component is composed of one or more classical deep learning models. These models process the most recent observations, yielding predictions for the physical unknowns of interest. The second block takes as inputs the physical variables, whether known or predicted by the neural networks, along with initial conditions. This second component time integrates one or multiple PDEs to generate the subsequent state of the system. The fourth order Runge–Kutta (RK4) method is used for time integration. These PDEs encode essential physical knowledge. As already discussed in the previous Sect. 2.4, the spatial derivatives are approximated using convolutional layers.

The parameters of the first component are trainable, they are optimised during training to estimate the unknown variables. However, we froze the parameters of the second block, as it represents already-known operations. This ensures that the second block maintains its fixed structure, representing the known physical knowledge encoded in the equations, while the trainable block focuses on learning and predicting the unknown aspects of the system. This architecture combines the physical knowledge encoded in the equations with the pattern-extraction capabilities of neural networks.

In the following part, we employ this architecture for cloud cover nowcasting, with different models being implemented, each using a different physical modelling approach.

## 3.2 Physical modelling

Before delving into the details of the proposed models, let us first establish the essential characteristics of the data at hand. In this work, we investigate cloud cover nowcasting over France, using cloud cover satellite images captured by the Meteosat Second Generation (MSG) satellite at 0 degrees longitude. The data's spatial resolution over France is $\approx$ 4.5 km and the time step is 15 minutes. These 256×256 satellite images have been processed by EUMETSAT (García-Pereda et al., 2019), classifying each pixel among sixteen distinct categories. We only considered cloud related categories, twelve in total.

In what follows, we introduce four models: HYPHAI-1 uses an advection equation to capture the motion of clouds. HY-PHAI-2 extends this by incorporating a simple source term in the advection equation. HYPHAI-3 uses a more complete but costly source term based on markovian inter-class transitions. Finally, HYPHAI-4 restricts the number of possible inter-class transitions to render HYPHAI-3 more tractable.

### 3.2.1 Advection equation: HyPhAI-1

To easily model the advection of these maps with different cloud types, we adopt a probabilistic approach, i.e. rather than representing a single map showing assigned labels, we use twelve maps, each representing the likelihood or probability of a specific cloud type being present at a given location. These maps must satisfy the following properties:

1. Non-negativity: $P(\mathbf{x},t) \geq 0$ for all $\mathbf{x}$ and $t$, with $\mathbf{x} = (x,y)$, which ensures that the probabilities remain non-negative.

2. Bound preservation: $P(\mathbf{x},t) \leq 1$ for all $\mathbf{x}$ and $t$, which ensures that no probability exceeds 1.





3. Probability conservation: $\sum_{i=1}^{C} P_X^i(\mathbf{x}, t) = 1$ for all $\mathbf{x}$ and $t$, with $C = 12$ is the total number of cloud types. This property guarantees that the sum of all probabilities is equal to 1.

This approach, known as "one-hot encoding", is more natural to address classification tasks. It involves using twelve distinct advection equations, each corresponding to a specific cloud type, as described below:

$$\partial_t P_j + \overrightarrow{V} \cdot \overrightarrow{\nabla} P_j = 0 \quad \forall j \in \{1, 2, \ldots, C\}, \tag{17}$$

where $P_j(\mathbf{x})$ represents the classification probability of the $j$-th cloud type, $\overrightarrow{V}(\mathbf{x})$ is the velocity field and has two component $u(\mathbf{x})$ and $v(\mathbf{x})$. Finally, $\overrightarrow{\nabla}$ denotes the gradient operator.

While one might initially perceive similarities between this modelling and a Fokker–Planck equation (Fokker, 1914; Pavliotis and Stuart, 2008, chap.6), the modelling approach presented here deviates significantly from the Fokker-Planck equation. In contrast, the Fokker-Planck equation is typically employed to depict the evolution of probability distributions for time continuous Markov processes over continuous states, e.g. Brownian motion. On the other hand, Eq. (17) characterises the probability advection for each finite state.

Nevertheless, by employing equations in the following form:

$$\partial_t P_j + \mathcal{L}(P_j) = 0 \quad \forall j \in \{1, 2, \ldots, C\}, \tag{18}$$

where $\mathcal{L}$ represents a differential operator with non-zero positive derivative orders, we have demonstrated in Appendix C that the probability conservation property is maintained over time. This assertion holds even in scenarios where the discretisation scheme introduces some diffusion or dispersion effects during the resolution process (see Appendix C2 and Appendix D). However, the non-negativity and bound preservation properties are compromised when a discretisation scheme with dispersion effects is used, unlike the diffusive schemes. Consequently, we opt for the first-order upwind diffusive discretisation scheme (see Appendix D2 for details about the equivalent equation) along with the RK4 for time integration. During the time integration process, we performed the integration by subdividing the time step $\Delta t = 1$ (representing 15 minutes) into smaller steps $\delta t = 0.1$ to check the Courant-Friedrichs-Lewy (CFL) condition (Courant et al., 1928), this condition ensures the stability of the numerical solution.

In the first hybrid model, denoted HYPHAI-1, we use a U-Net Xception-style model (Tamvakis et al., 2022) inspired from Xception architecture (Chollet, 2017). It takes previous observations, and estimates the velocity field (see Fig. 3). This model will be guided during training by the advection equation to learn the cloud motion patterns.

### 3.2.2 Advection with source term: HyPhAI-2

As the advection alone doesn't take into account other physical processes, especially, class change, appearance and disappearance of clouds, we propose to add a trainable source term to capture them. In this first attempt, we use a simple source term:

$$\partial_t P_j = \tanh(S_j) \quad \forall j \in \{1, 2, \ldots, C\}, \tag{19}$$





where $S_j$ is a 2D map. The hyperbolic tangent activation function ($\tanh$) is used to keep the values of the source term in a range of $[-1, 1]$, preventing it from exploding.

The second version of the hybrid model, denoted HYPHAI-2, adds this source term to the advection. This modelling is
described in the following equations:

$$\partial_t P_j + \overrightarrow{V} \cdot \overrightarrow{\nabla} P_j = \tanh(S_j) \quad \forall j \in \{1, 2, \ldots, C\}, \tag{20}$$

where $S_j$ is estimated using a second U-Net model (see Fig. 4).

### 3.2.3 Advection with source term: HyPhAI-3

While the previous modelling describes the missing physical process in the advection, it doesn't satisfy the probability con-
servation property. Thus, this modelling does not conserve the probabilistic nature of $P$ over time. To ensure the appropriate
dynamics of probability, a robust framework is provided by continuous-time Markov processes across finite states (Pavliotis
and Stuart, 2008, chap. 5). In this framework, the probability trend is controlled by a linear dynamics, ensuring the bound
preservation, positivity and probability conservation. This dynamics is expressed using the following equations:

$$\partial_t P_j = \sum_{i=1}^{C} \boldsymbol{\Lambda}_{j,i} P_i \quad \forall j \in \{1, 2, \ldots, C\},$$

where

$$\Lambda(\mathbf{x}) = \frac{\boldsymbol{\Pi}(\mathbf{x})^{\mathsf{T}} - I}{\Delta t},$$

with $\boldsymbol{\Pi}(\mathbf{x})$ being a stochastic matrix, i.e. a non-negative square matrix where the sum of each row is equal to one. This
constraint ensures that the probabilistic properties are maintained over time.

Physically, $\boldsymbol{\Lambda}_{j,i}(\mathbf{x})$ represents the transition rate from cloud type $i$ to cloud type $j$ at grid point $\mathbf{x}$ and $\Delta t$ represents the time
step, and $I(\mathbf{x})$ denotes the identity matrix.

The third version of the hybrid model (see Fig. A1), denoted HYPHAI-3, uses this source term combined with the advection
as showed in the following equations:

$$\partial_t P_j + \overrightarrow{V} \cdot \overrightarrow{\nabla} P_j = \sum_{i=1}^{C} \boldsymbol{\Lambda}_{j,i} P_i \quad \forall j \in \{1, 2, \ldots, C\}, \tag{21}$$

where the stochastic property of $\boldsymbol{\Pi}$ is ensured by construction using the *Softmax* function as follows:

$$\Pi_{i,k} = \text{Softmax}(M_i)_k = \frac{e^{M_{i,k}}}{\sum_{j=1}^{C} e^{M_{i,j}}},$$

where the matrix $M$ is generated using a U-Net.

This representation of cloud cover dynamics offers a comprehensive description of cloud formation and dissipation. How-
ever, it increases the output dimension size of the U-Net, as a $C^2$ transition matrix is generated for each pixel. This makes the
U-Net model poorly constrained. Furthermore, in our experiments, we noticed an increased memory usage during the training.



### 3.2.4 Advection with source term: HyPhAI-4

To reduce the number of values output by the U-Net, we assume a limited number of transition regimes, each representing one of the most frequent transitions. For instance, in the case of two regimes, the source term is described as follows:

$$\partial_t P_j = \alpha^1 \cdot \sum_{i=1}^{C} \Lambda_{j,i}^1 P^i + \alpha^2 \cdot \sum_{i=1}^{C} \Lambda_{j,i}^2 P^i,$$

where $\Lambda^1$ and $\Lambda^2$ are the transition matrices, $\alpha^1$ and $\alpha^2$ are positive factors, these factors determine which regime to consider at each pixel, with the constraint that $\alpha^1 + \alpha^2 \leq 1$.

The fourth version of the hybrid model, denoted HYPHAI-4, uses this source term in addition to the advection as described in the following equations:

$$\partial_t P_j + \overrightarrow{V} \cdot \overrightarrow{\nabla P_j} = \alpha^1 \cdot \sum_{i=1}^{C} \Lambda_{j,i}^1 P^i + \alpha^2 \cdot \sum_{i=1}^{C} \Lambda_{j,i}^2 P^i, \tag{22}$$

where, $\alpha^1$ and $\alpha^2$ are estimated using a U-Net, and $\Lambda^1$ and $\Lambda^2$ are learned as model parameters (see Fig. A2).

### 3.3 Training procedure

The training was carried out on a dataset containing about three years of data from 2017 to 2019, with a total of approximately 100,000 images. To improve the diversity of the training set and take into account a possible overfitting on the typical movements of clouds in the Western Europe region, we randomly applied simple transformations to the images, more precisely, rotations of 90, 180 and 270 degrees, which increased the dataset size and improved the model's ability to learn various cloud motion patterns. After cleaning, about 8000 sequences of 12 images were used for training and about 400 sequences for validation. The test set was done on a separate dataset from the same region but from the year 2021.

We used the PyTorch framework to implement the models, and we employed the cross-entropy loss function for training. This function is given by:

$$l(Y,p) = -\frac{1}{N} \sum_{i=1}^{N} \sum_{j=1}^{C} Y_{i,j} \log(p_{i,j}), \tag{23}$$

where $N$ represents the total number of pixels, $C$ denotes the number of classes, $p_{i,j}$ is the predicted probability of the $i$-th pixel belonging to the $j$-th class, and $Y_i$ corresponds to the one-hot encoded ground truth at the $i$-th pixel, i.e. $Y_{i,j} = 0$ except for the correspondent cloud type, where $Y_{i,j} = 1$.

The training of the model parameters is achieved through gradient-based methods, which rely on computing the loss gradients with respect to the model parameters. These gradients guide the update of the model's weights during the training process. Here, Adam optimiser (Kingma and Ba, 2017) is used with a learning rate of $10^{-3}$ and a batch size of 4 with 16 accumulation steps, which allows us to simulate a batch size of 64. The training was performed using a single Nvidia A100 GPU for 30 epochs.

You can find the source code for our project on GitHub at https://github.com/relmonta/hyphai.





## 3.4 Performance metrics

To evaluate the performance of competing models in this study, we employed various metrics. Firstly, standard classification metrics are used to evaluate the statistical aspect, then the Hausdorff distance is introduced to evaluate the qualitative aspect of the results.

### 3.4.1 Classic classification metrics

The selected metrics include Accuracy, Precision, Recall, F1 score, and the Critical Success Index, or CSI (Gilbert, 1884), also
called Intersection over Union (IoU) or Jaccard Index. These metrics offer multiple insights into different aspects of model performance. Accuracy measures the proportion of correct predictions, while Precision quantifies the proportion of correct positive predictions relative to the total number of positive predictions. Recall evaluates the proportion of correct positive predictions relative to the total number of positive cases. The F1 score provides a balance between Precision and Recall. The CSI measures the overlap between the prediction and ground truth, providing a measure of similarity.

To compute these metrics for the $j$-th class, we use the following formulas:

$$\text{Accuracy}_j = \frac{\text{TP}_j + \text{TN}_j}{\text{TP}_j + \text{TN}_j + \text{FP}_j + \text{FN}_j},$$

$$\text{Recall}_j = \frac{\text{TP}_j}{\text{TP}_j + \text{FN}_j},$$

$$\text{Precision}_j = \frac{\text{TP}_j}{\text{TP}_j + \text{FP}_j},$$

$$\text{F1}_j = \frac{2 \times \text{Precision}_j \times \text{Recall}_j}{\text{Precision}_j + \text{Recall}_j},$$

$$\text{CSI}_j = \frac{\text{TP}_j}{\text{TP}_j + \text{FP}_j + \text{FN}_j}.$$

These metrics are calculated separately for each class, where TP denotes instances correctly identified as positive cases, TN refers to instances correctly identified as negative cases, FP represents cases miss-classified as positives, and FN is the number of positive cases that are classified as negative.

To obtain an overall performance evaluation of the Accuracy, we use the following formula:

$$\text{Accuracy} = \frac{\sum_j \text{TP}_j}{\text{Total number of cases}}.$$

For the remaining metrics, we can calculate two types of average: the macro-average and the micro-average. The macro-average is the arithmetic mean of the metric scores computed for each class, while the micro-average considers all classes as a single entity (Takahashi et al., 2022). Given the highly imbalanced distribution of labels in our dataset, we adopted the macro-average to evaluate the models' performances (Fernandes et al., 2020; Wang et al., 2021). The macro-averaged F1 is defined as in Sokolova and Lapalme (2009) as follows:

$$\text{F1}_{macro} = \frac{2 \times \text{Precision}_{macro} \times \text{Recall}_{macro}}{\text{Precision}_{macro} + \text{Recall}_{macro}},$$





where the macro-averaged Precision and Recall are defined as:

$$\text{Precision}_{macro} = \frac{1}{C} \sum_{j=1}^{C} \text{Precision}_j.$$

$$\text{Recall}_{macro} = \frac{1}{C} \sum_{j=1}^{C} \text{Recall}_j.$$

We define the macro-averaged CSI following the same method as follows:

$$\text{CSI}_{macro} = \frac{1}{C} \sum_{j=1}^{C} \text{CSI}_j.$$

These pixel-wise metrics are commonly used for evaluating image segmentation tasks or more generally classification tasks, but it is important to note the limitations of these metrics and evaluation approaches. While the selected metrics provide valuable insights, they do not capture all aspects of model performance, for instance, because they do not take into account the spatial

correspondence between predicted and ground-truth cloud structures. This means that a model can statistically perform well using pixel-wise metrics, but still have poor performance in identifying the correct cloud structures or missing a significant amount of detail. As a result, evaluating cloud cover forecasting models based solely on pixel-wise metrics may not be sufficient to ensure their effectiveness in real-world applications.

### 3.4.2 Hausdorff distance

The Hausdorff distance is a widely used metric for medical image segmentation (e.g. Karimi and Salcudean, 2019; Aydin et al., 2021), this metric measures the similarity between the predicted region and the ground truth region, by comparing structures, rather than just individual pixels. It can be expressed using either Eq. (24) or Eq. (25) described as follows:

$$h^1(A,B) = \frac{1}{|A|} \sum_{p \in A} \min_{q \in B} d(p,q), \tag{24}$$

$$h^2(A,B) = \max_{p \in A} \min_{q \in B} d(p,q), \tag{25}$$

where $d(p,q)$ is the Euclidean distance between $p$ and $q$. The former computes the mean distance between each point $A$ and the closest point in $B$, providing an overall measure of similarity. The latter measures the maximum distance between a point in $A$ and the closest point in $B$ (Fig. 5), this formulation is a more conservative measure that focuses on the largest discrepancies between the sets. Both formulations exhibit sensitivity to the loss of small structures. Specifically, when small regions in

the ground truth are non-empty while their corresponding regions in the prediction are empty, the search area expands, which increases the overall distance. We opt to limit this search region to the maximum distance traversable by a cloud. Consequently, we introduce the *restricted Hausdorff distance* (rHD) defined as follows:

$$h^3(A,B) = \frac{1}{|A|} \sum_{p \in A} \min_{q \in \mathbf{B}_r(p)} d(p,q), \tag{26}$$





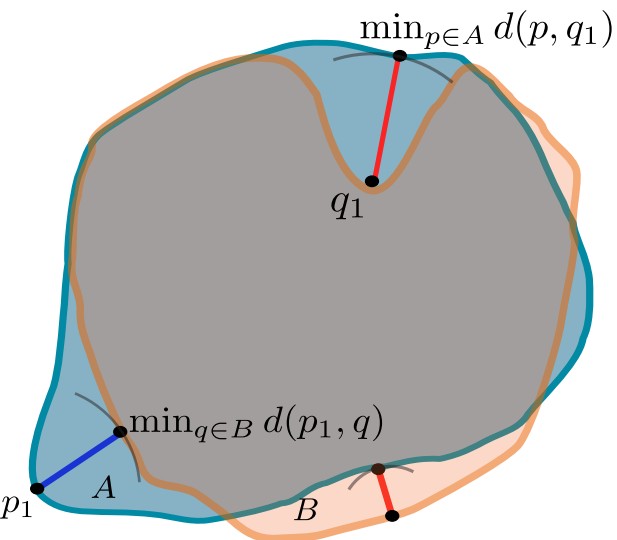

**Figure 5.** Illustration of the $\min_{p \in A} d(p, q_1)$ and $\min_{q \in B} d(p_1, q)$ quantities used to compute the Hausdorff distance; for each point, we look for the closest point in the other region.

where $\mathbf{B}_r(p)$ is the ball of radius $r$ centred at $p$. In our experiments, we set $r$ to 10 pixels, which corresponds to a radius of
395 approximately 45-50 $km$, corresponding to the maximum distance crossed by clouds in one time step, considering 200 $km$ $h^{-1}$ as the cloud's maximum speed. This means that for each pixel in the first set, we compute the distance to the closest pixel in the second set, but only if it is within a radius of 10 pixels. This allows us to reduce the impact of small regions in the ground truth that are not present in the prediction, while still rewarding the model if it correctly predicts them.

The Hausdorff distance is a directed metric, i.e. $h^p(A, B) \neq h^p(B, A)$, thus, we consider the maximum of the two directed
distances as follows:

$$\mathcal{H}(S, \hat{S}) = \max\left(h^3(S, \hat{S}), h^3(\hat{S}, S)\right) \tag{27}$$

where $S$ and $\hat{S}$ are the coordinates of positive pixels in the ground truth and prediction, respectively.

### 3.5 Benchmarking procedure

To assess the performance of the proposed models, we consider established benchmarks. In the comparative evaluation, we
included the widely used U-Net (Ayzel et al., 2020; Berthomier et al., 2020; Trebing et al., 2021, e.g.). U-Net architecture is structured with a contracting path and an expansive path, connected by a bottleneck layer. The contracting path comprises four levels of convolutional layers, each followed by a max-pooling layer. The number of filters we used in these convolutional layers progressively increases from 32 to 64, 128, and finally 256. On the other hand, the expansive path consists of four sets



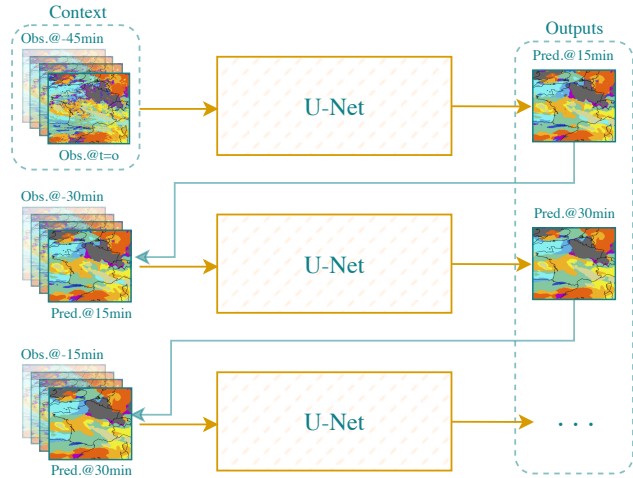

**Figure 6.** The U-Net architecture considered in the comparison. The U-Net is iteratively used to predict the next state given the previous ones.

of convolutional layers, each followed by an upsampling layer. These layers help in the reconstruction and expansion of the
feature maps to match the original input size. We iterate over the U-Net, as illustrated in Fig. 6, to generate predictions for
multiple future time steps.

In addition to U-Net, we consider in our comparison, a product based on kinematic extrapolation techniques, called EXIM
(for Extrapolated Imagery), developed by EUMETSAT as part of their NWCSAF/GEO products (García-Pereda et al., 2019).

We also included a commonly used meteorological baseline method known as "Persistence". This method predicts future
time steps by simply using the last observation, a relevant approach in nowcasting, since weather changes occur slowly, making
the last observation a strong prediction, which makes the Persistence baseline challenging to outperform.

We tested the competing models using 1000 satellite images samples captured over France from January 2021 to October
2021.

## 4 Experiments and results

We trained the hybrid models, in addition to the U-Net used for comparison, on three years of data. The models were designed
to predict a 2-hour forecast at 15-minute intervals.

In what follows, our attention will be directed only towards HyPhAI-1 and HyPhAI-2. This choice is made as HyPhAI-3 and
HyPhAI-4 demonstrated performance levels identical to that of HyPhAI-1. Indeed, the $\Lambda$ matrices in Eq. (21) and Eq. (22) are
consistently estimated as zeros. In other words, no inter-class transitions were captured.





**Figure 7. Performance comparison between our HyPhAI-1, U-Net, EXIM, and the Persistence baseline**. Using five metrics including averaged F1 score(%), precision(%), recall(%), accuracy(%), CSI(%) and Hausdorff distance (defined in Eq. (27)). These scores were computed over 1000 random samples covering France in 2021. The confidence intervals were estimated using Bootstrapping.

Table 1. Score comparison at the 120-minute lead time (↑: higher is better, ↓: lower is better)

| Model | ↑ F1 score | ↑ Precision | ↑ Recall | ↑Accuracy | ↑ CSI | ↓ Hausdorff distance ($\mathcal{H}$) |
|---|---|---|---|---|---|---|
| HYPHAI-1 | **26.6 %** | **27.5 %** | **25.9 %** | 55.4 % | **17.2 %** | 6.23 |
| HYPHAI-2 | **26.5 %** | **27.6 %** | 25.7 % | **57.3 %** | **17.1 %** | 6.54 |
| U-Net | 24.9 % | 25.6 % | 24.5 % | 56.0 % | 16.1 % | 6.90 |
| EXIM | 23.5 % | 23.5 % | 23.6 % | 49.4 % | 14.9 % | **5.08** |
| Persistence | 21.8 % | 21.9 % | 21.8 % | 47.9 % | 13.8 % | 5.53 |



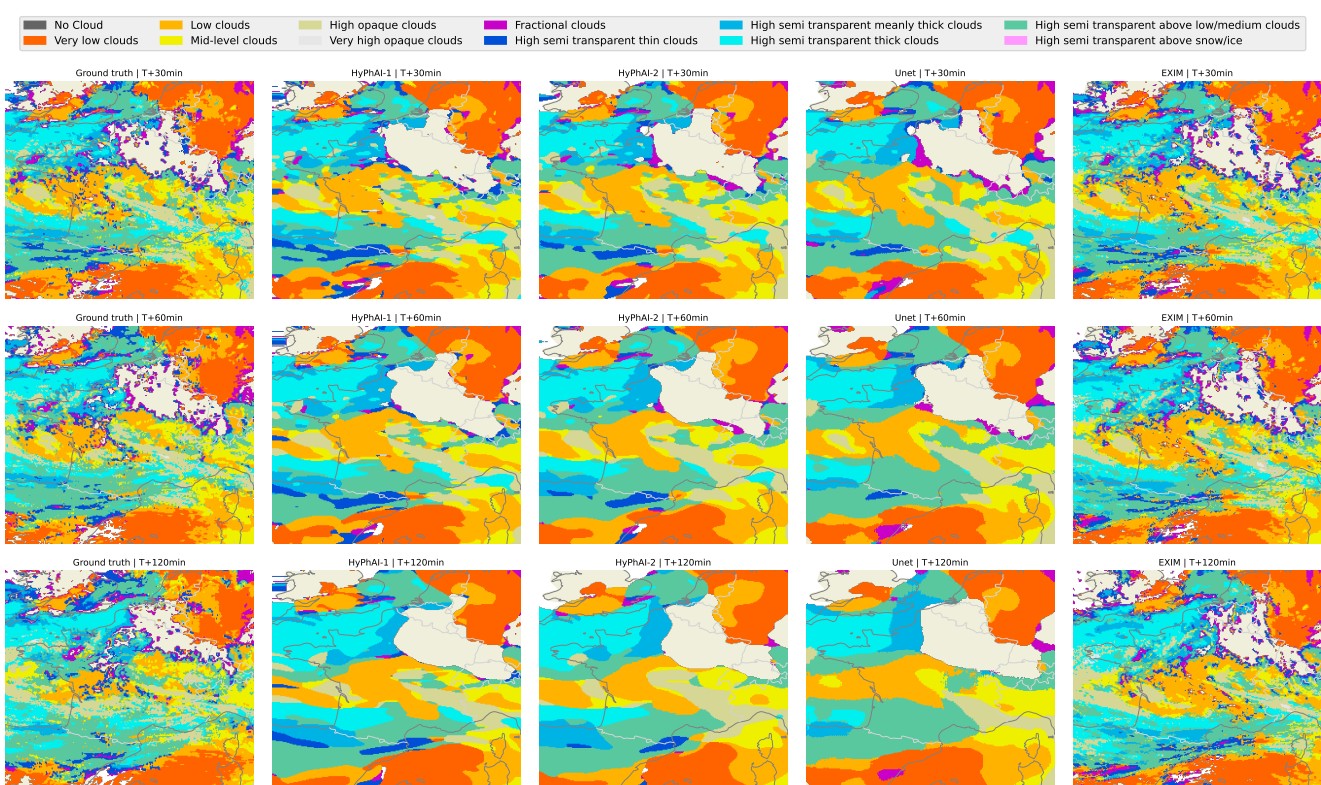

**Figure 8. Case study of different models' forecasts**. Left column: ground truth at different time steps; middle columns: HyPhAI-1, HyPhAI-2 and the U-Net's predictions, respectively; right column: EXIM's predictions.

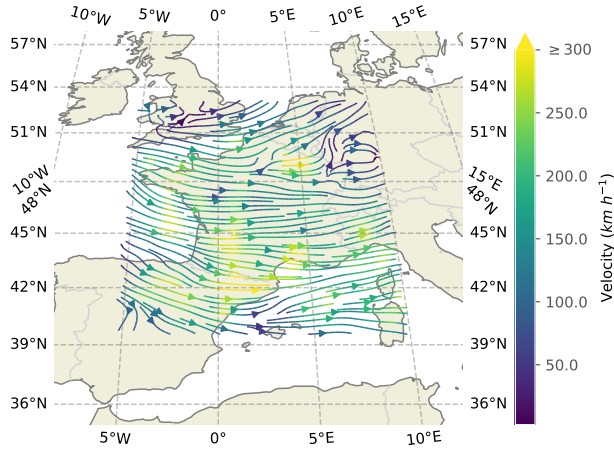

**Figure 9.** Estimated velocity field by the U-Net Xception-style used in the HyPhAI-1 model



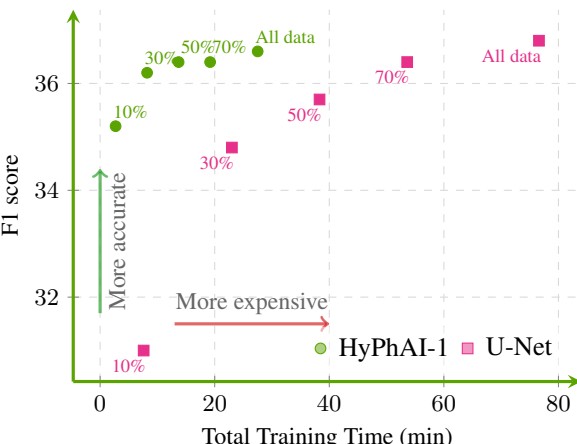

**Figure 10.** Total training time and maximum validation F1 scores over the last 5 epochs for the U-Net and HyPhAI-1 using different training data sizes (averaged over all the lead times).

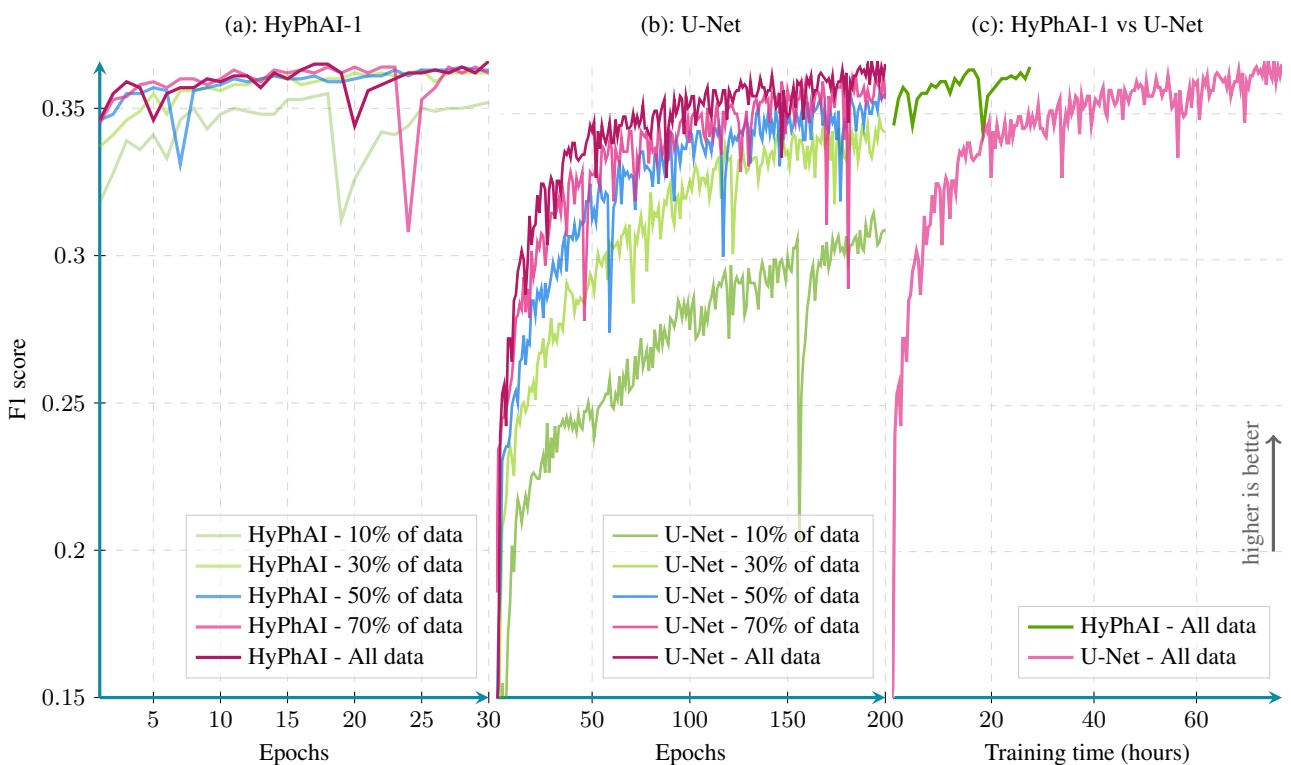

**Figure 11. Per epoch validation F1 score comparison between HyPhAI-1 and the U-Net**. Scores were calculated from 100 random samples covering France (averaged over all the leadtimes).





## 4.1 Quantitative analysis

Diving into the numerical evaluations, here we present a comparative analysis based on standard metrics used in image classification tasks. Figure 7 shows a score comparison using different metrics over multiple lead time, up to two hours. The confidence intervals, indicating statistical significance, are computed using a resampling method, called *bootstrap*, which is a statistical technique that involves repeatedly sampling from a single dataset to generate numerous simulated samples (Efron, 1979). Through this method, standard errors, confidence intervals, and hypothesis testing can be computed. Table 1 and Fig. 7 show that HyPhAI-2 is slightly better in terms of precision and accuracy than the model using advection equation without source term (HyPhAI-1), and both of these hybrid models significantly outperform the U-Net in terms of F1 score, precision, and CSI, and perform similarly in terms of accuracy and recall. This is because the U-Net tends to give more weight to the dominant classes at the expense of the other classes, resulting in a higher false positive rate.

While quantitative performance metrics offer a numerical assessment of a model's ability to predict weather states, providing crucial insights into the reliability and precision of forecasts, they are not sufficient on their own. Qualitative aspects also play a significant role, including the visual interpretation of model predictions and an assessment of its capability to capture complex atmospheric patterns and phenomena.

## 4.2 Qualitative analysis

Figure 8 presents a case study involving multiple models, highlighting that HyPhAI-1 produces more realistic and less blurry forecasts compared to the U-Net. To substantiate this claim, we used the restricted Hausdorff distance (rHD), described in Eq. (26), to assess the sharpness of predicted cloud boundaries. Both models HyPhAI-1, HyPhAI-2 outperformed the U-Net in this metric, as shown in Fig. 7. EXIM and the Persistence baseline exhibit superior results in terms of the Hausdorff metric, and the gap between them and the other models increases with the lead time, which is visually expected. The reason behind this result is that the hybrid models, especially HyPhAI-1, preserve more details compared to the U-Net, the lost details in HyPhAI-1's predictions are only due to the diffusion added numerically by the discretisation scheme used (refer to D for more details). Whereas the U-Net focuses more on dominant structures and labels, which are more likely to persist over time, which is statistically relevant. Nonetheless, EXIM and the Persistence baseline still outperform the other models in this regard. This observation aligns with the fact that the Persistence uses the last observation as its predictions, and EXIM is advecting the last observation while keeping the same level of details. However, EXIM is slightly more accurate, compared to Persistence, in terms of predicted cloud positions.

In Figure 9, we present the estimated velocity field generated by the HyPhAI-1 model, illustrated in Fig. 3. This field exhibits a high level of coherence with the observed cloud cover displacements, with exceptions in cloud-free areas, as expected. It is important to emphasize that this velocity field is derived exclusively from cloud cover images, without relying on external wind data or similar sources. This aspect adds a layer of interest, especially in the context of other applications beyond the cloud cover nowcasting.



### 4.3 Time efficiency

In what follows, we focus only on the HyPhAI-1 model. By including physical constraints into these hybrid models, we expect a decrease in training time compared to that of the U-Net. Indeed, Fig. 11 illustrates the evolution of the validation F1 score for both the U-Net and the HyPhAI-1 model across epochs. HyPhAI-1 converges faster than the U-Net, indeed, its convergence occurs after just about 10–15 epochs. Each epoch of the HyPhAI-1 training takes approximately 55 minutes using a single Nvidia A100 GPU, the entire training over 30 epochs takes 27h. On the other hand, the U-Net necessitates up to 200 epochs for achieving similar performance, with each epoch taking around 23 minutes using the same hardware, which corresponds to thereabout three days of training. This difference implies that training the U-Net is significantly more expensive compared to the HyPhAI-1.

In inference mode, the HyPhAI models and the U-Net generate predictions within a few seconds, while EXIM's predictions are produced within 20 minutes (Berthomier et al., 2020), which is one of the main drawbacks of this product.

### 4.4 Data efficiency

To delve deeper into the efficiency of the proposed HyPhAI-1 model, we conducted various experiments using different training data sizes. In each experiment, both HyPhAI-1 and the U-Net were trained with 70 %, 50 %, 30 % and 10 % of the available training data (Fig. 11, Fig. 10). Notably, we observed a more significant decline in performance for the U-Net compared to HyPhAI-1. Interestingly, the hybrid model exhibited similar performance with only 30 % of the training data as it did with the entire dataset (Fig. 11). This finding indicates that this hybrid model is remarkably data-efficient, capable of delivering satisfactory performance even with limited training data. This quality is very important, particularly for tasks with insufficient provided data.

### 4.5 Application on earth's full disk

To check HyPhAI-1's capabilities on broader scales after training it on a small region, we tested it on a much larger domain, the full earth's disk centred at 0 degrees longitude. This expansive full disk domain is 14 times the size of the training area. It has diverse meteorological conditions and includes projection deformations when mapped onto a two-dimensional plane. Therefore, it provides an ideal testing ground for HyPhAI-1's generalisation ability. Despite the significant differences between the training domain and the full disk, we observed a remarkable adaptation of the HyPhAI-1 model to this new context without any specific training on it (see Fig. A3). The cloud motion estimation on the full disk was found to be accurate and reliable, this successful transferability of the model highlights its robustness and suggests that the underlying principles of cloud motion captured during training are applicable across different domain sizes, and different projections (see Appendix B for a formal explanation). Note that the model requires a data size divisible by $2^d$, where $d$ is the number of the encoder blocks within the U-Net-Xception model. Indeed, the possibility to run a model using different data sizes is one of the advantages of Fully Convolutional Networks (FCN) as the convolution operation is independent of the input size.



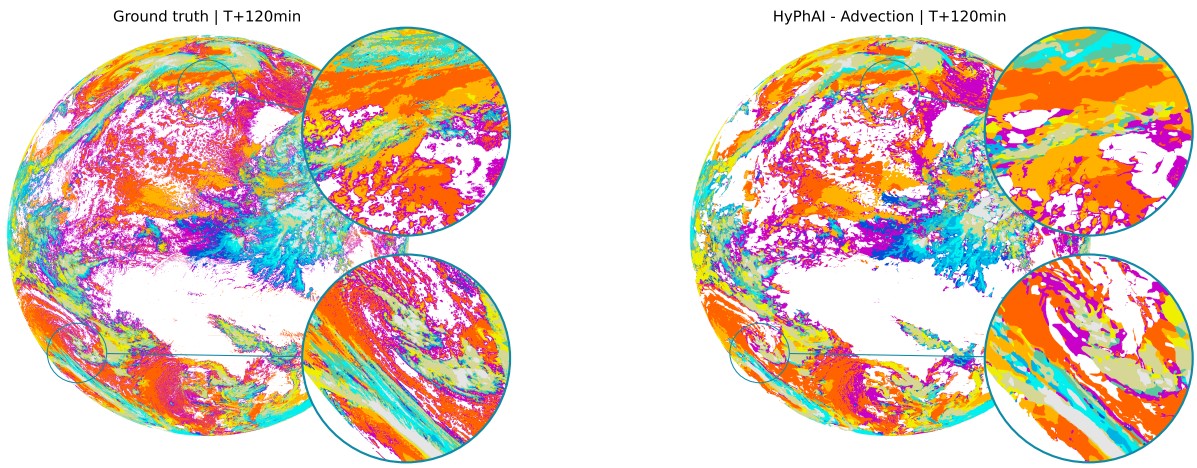

**Figure 12. Full disk cloud cover nowcasting predictions**. Zoomed-in views of the 120-minute observation and prediction.

Overall, HyPhAI-1 offers an effective and cheaper approach compared to EXIM, with higher efficiency, requiring fewer data compared to the U-Net, with the potential to outperform existing models and enable more accurate and efficient weather forecasting. The ability of the HyPhAI-1 to adapt and perform well on the full-disk data, despite being trained on a smaller domain, demonstrates the generalisation capabilities of this hybrid model. This is an important property for weather forecasting models, as it is not always possible to train a model on full-disk data due to the high computational cost.

## 5   Conclusions

In this study, we introduced a hybrid Physics-AI framework that combines the insights from partial differential equations, representing physical knowledge, with the pattern-extraction capabilities of neural networks. Our primary focus was on applying this hybrid approach to the task of cloud cover nowcasting, also involving cloud type classification. To leverage continuous physical advection phenomena for this discrete classification task, we proposed a probabilistic modelling strategy based on the advection of probability maps. This flexible approach was easy to adapt to include the prediction of source terms, demonstrating its versatility.

The first model, HyPhAI-1, leverages the advection equation and slightly outperforms the widely used U-Net in the quantitative metrics, while showing a significantly better performance in the qualitative aspect. This hybrid model requires significantly less amount of data and converges faster, cutting down the training time, which is expected as the physical modelling implicitly imposes a constraint on the trainable component. Notably, the estimated velocity field demonstrated high accuracy compared to actual cloud displacements. This accuracy suggests that this architecture could find utility in diverse tasks, such as wind speed estimation using only satellite observations. The second version, HyPhAI-2, adds a source term to the advection equation. This model impaired the visual rendering but displayed the best performance in terms of Accuracy.



The HyPhAI architecture demonstrated an effective path towards uniting the strengths of a continuous physics-informed phenomenon with a data-driven approach, in the context of a discrete classification task.

Despite these successes, the models still exhibit some diffusiveness. However, in the case of HyPhAI-1, it is only attributed to the use of the first-order upwind discretisation scheme. Exploring less diffusive schemes could potentially mitigate this effect, especially in inference mode, where there is no differentiability constraint.

The CFL condition is designed to guarantee stability by imposing a restriction on the time step size relative to the maximum velocity in the system. However, in our scenario, where the maximum velocity of the cloud is unknown, setting the time step becomes challenging. This uncertainty may lead to stability issues if the time step is not small enough, particularly if the predicted velocity turns out to be unexpectedly high, highlighting the importance of carefully considering and addressing potential instability concerns in such cases.

While HyPhAI-3 and HyPhAI-4 presented interesting modelling variations, the study acknowledges limitations in not obtaining the desired variables. We suggest that modifying the approach to estimate these variables may lead to improved results, e.g. penalising the dominant classes.

As we move forward, the integration of green computing principles into AI research becomes crucial. The success of the HyPhAI models in achieving these results with low data requirement and rapid convergence encourages further exploration of energy-efficient AI models and methodologies. This emphasises the importance of balancing computational power with environmental responsibility in the pursuit of scientific advancements.

*Code and data availability.* The code used in this study is available at https://github.com/relmonta/hyphai (last access: 18 February 2024) and at https://doi.org/10.5281/zenodo.10676679. Weights of the pre-trained HyPhAI-1, HyPhAI-2 and the U-Net are available at https://doi.org/10.5281/zenodo.10393415. The training data is not provided as it is proprietary data from EUMETSAT. However, the data can be obtained from EUMETSAT for research purposes. A sample of the test data used in this study is available on the GitHub repository, and a sample of the training data is available at https://doi.org/10.5281/zenodo.10642094.

*Video supplement.* A video supplement of a 2-hour forecast is available at https://doi.org/10.5281/zenodo.10375284.

## Appendix A: Additional resources

### A1 Architectures

The HyPhAI-3 model using transitions matrices as a source term (described in Eq. (21)) is shown in Fig. A1. In the Fig. A2, we show a diagram presenting the HyPhAI-4 model (described in Eq. (22)), which is using a limited number of transitions regimes.



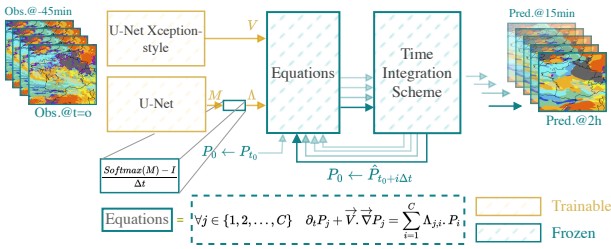

**Figure A1.** HYPHAI-3: The third version of the proposed hybrid model. It consists of a U-Net Xception-style to estimate the velocity field and a second U-Net to estimate the per-pixel transition matrices from the last observations

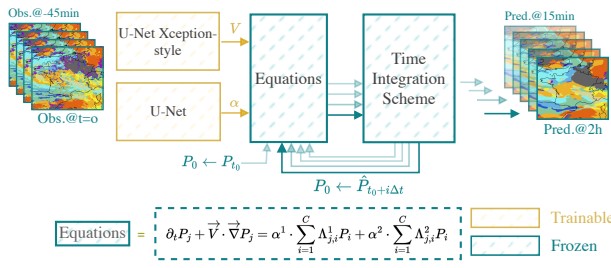

**Figure A2.** HYPHAI-4: The fourth version of the proposed hybrid model. It consists of a U-Net Xception-style to estimate the velocity field and a second U-Net to estimate the $\alpha$ factors from the last observations, these factors are used to choose which transition regime to consider for each pixel.

## A2 Full disk predictions

The Fig. A3 shows predictions of the HyPhAI-1 model on the earth full disk centred at 0 degrees longitude.

## Appendix B: Robustness of hybrid formulation to change of coordinates

In a given coordinate system $x = (x_i)$, the advection of a passive scalar $c(t,x)$ by a velocity field $u = (u_i)$ reads as

$$\partial_t c + u_i \partial_{x_i} c = 0. \tag{B1}$$

A change of coordinate system from the coordinate system $x$ to the coordinate system $y = (y_j)$ related by $x = x(y)$, remains to the dynamics

$$\partial_t C + v_j \partial_{y_j} C = 0, \tag{B2}$$

where $C(t,y) = c(t,x(y))$ and where the velocity $v = (v_j)$ is deduced from the chain rule

$$v_j = u_i \partial_{x_i} y_j, \tag{B3}$$





**Figure A3. Full disk cloud cover nowcasting predictions**. The predictions were generated by our model without any specific training on the full disk data (of size $3712 \times 3712$).



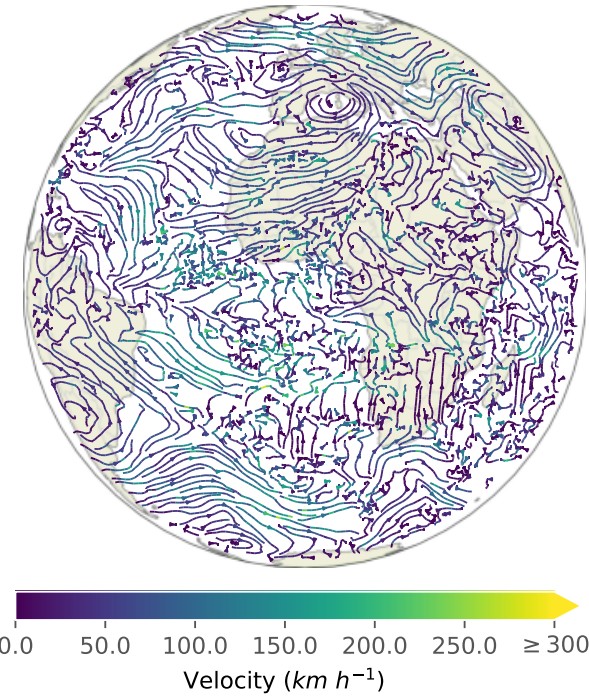

**Figure B1.** Estimated velocity field by the U-Net Xception-style used in the HyPhAI-1 model.

(using Einstein's summation convention).

Since HyPhAI architecture estimates a velocity field from the data, that is either $u$ or $v$, depending on the choice of the coordinate system, it implicitly accounts for the chain rule Eq. (B3). As a result, the HyPhAI architecture is not sensitive to the coordinate system and can apply to regional domain as well as global projections. However, numerical effects due to the finite spatio-temporal resolution associated with the discretisation, can lead to abnormal distortion of signals after several time steps

of integration, e.g. the disk resulting from an orthographic projection of the Earth may be deformed by the advection near its boundaries unless the velocity field is close to zero, meaning that the apparent displacement is small.

Note that this relative invariance of HyPhAI to the choice of coordinate is because it only concerns the advection of a scalar field. Covariant transport of vector or tensor fields would imply additional terms (Christoffel symbols, e.g. Nakahara (2003)) that would break the invariance of HyPhAI as it is formulated here.

**Appendix C: Probability advection**

In this section, we are considering a three-class problem where we have a discrete random variable $X$ with values in the set $1, 2, 3$, and we denote by $X(t, x)$ the value of $X$ at time $t$ and space $x$, with $t \in [0, T]$ and $x \in [0, L]$. We are interested in





studying the evolution of the state probabilities of $X$ with respect to $t$ and $x$. For this purpose, we define a vector $\mathcal{P}$ as

$$
\mathcal{P} = \left[ \begin{array}{c} P_X^1 \\ P_X^2 \\ P_X^3 \end{array} \right],
$$

here, $P_X^c(t,x)$ represents the probability of the $c$-th class;

$$
P_X^c(t,x) = P(X(t,x) = c)
$$

.

For the sake of simplicity, a 1D problem is considered, but the same analysis applies to the 2D case and for $N$-class problems with $N \geq 2$. Let's consider the following partial differential equation governing the evolution of $\mathcal{P}(x,t)$:

$$\partial_t \mathcal{P}(x,t) + \mathcal{L}(\mathcal{P}(x,t)) = 0, \tag{C1}$$

where $\mathcal{L}$ is a differential operator. This equation can be written component-wise as:

$$
\begin{cases} \partial_t P_X^1(x,t) + \mathcal{L}(P_X^1(x,t)) = 0 \\ \partial_t P_X^2(x,t) + \mathcal{L}(P_X^2(x,t)) = 0 \\ \partial_t P_X^3(x,t) + \mathcal{L}(P_X^3(x,t)) = 0 \end{cases} \tag{C2}
$$

As already discussed in the Sect. 3.2.1, three properties should be checked in order to ensure the probabilistic nature of $P$.

1. Non-negativity: $P(\mathbf{x},t) \geq 0$ for all $\mathbf{x}$ and $t$, with $\mathbf{x} = (x,y)$, which ensures that the probabilities remain non-negative.

2. Bound preservation: $P(\mathbf{x},t) \leq 1$ for all $\mathbf{x}$ and $t$, which ensures that no probability exceeds 1.

3. Probability conservation: $\sum_{i=1}^{C} P_X^i(\mathbf{x},t) = 1$ for all $\mathbf{x}$ and $t$, with $C = 12$ is the total number of cloud types. This property guarantees that the sum of all probabilities is equal to 1.

## C1  Probability conservation

**Property .**The probability conservation property is ensured if $\mathcal{L}$ is a linear differential operator with non-zero positive spatial
derivative orders.

*Proof.* Let's sum the three equations in Eq. (C2):

$$
\sum_{i=1}^{3} \partial_t P_X^i(x,t) + \mathcal{L}(P_X^i(x,t)) = 0
$$

$$
\partial_t \sum_{i=1}^{3} P_X^i(x,t) = -\sum_{i}^{3} \mathcal{L}(P_X^i(x,t)),
$$





For the specific case where $\mathcal{L}$ is a linear differential operator with non-zero positive spatial derivative orders. Assuming $\sum_{i=1}^{3} P_X^i(x,t_0) = 1$, the linearity property of $\mathcal{L}$ allows us to interchange the summation and the operator, resulting as follows:

$$\sum_i^3 \mathcal{L}\left(P_X^i(x,t_0)\right) = -\mathcal{L}\left(\sum_{i=1}^3 P_X^i(x,t_0)\right)$$
$$= -\mathcal{L}\left(1\right)$$
$$= 0$$

$\mathcal{L}\left(1\right) = 0$ as $\mathcal{L}$ have only derivatives with positive non-zero orders.

Applying and summing the first order Taylor expansion at $t_0$ on each of the time derivatives of Eq. (C2) give

$$\sum_i^3 \frac{P_i(x,t_0 + \delta t) - P_i(x,t_0)}{\delta t} + \mathcal{O}(1) = -\sum_i^3 \mathcal{L}\left(P_X^i(x,t)\right)$$
$$= 0$$

$$\sum_i^3 P_i(x,t_0 + \delta t) = \sum_i^3 P_i(x,t_0) + \mathcal{O}(\delta t),$$

when $\delta t$ is small enough, $\sum_i^3 P_i(x,t_0 + \delta t) = 1$.

Iteratively, starting from $t_0$, $\forall t$

$$\sum_i^3 P_i(x,t) = \sum_i^3 P_i(x,t_0) = 1$$

$\square$

In this study, we consider the advection equation using the same velocity field for all probability maps, where the operator $\mathcal{L}$ is written as follows:

$$\mathcal{L}\left(P_i\right) = u \cdot \partial_x P_i, \quad i \in \{1,2,\cdot,12\}.$$

This differential operator is linear and have non-zero positive derivative order. Therefore, the sum of probabilities is conserved over time and remains equal to the initial value. This property is illustrated numerically in Fig. C2 and Fig. C4, and it is maintained independently of the discretization scheme.

## C2 Non-negativity and bound preservation

In order to check the two other properties, we need to study the discretisation schemes.

Out of the four numerical schemes studied (central finite differences, Semi-Lagrangian, first and second order upwind), only the Semi-Lagrangian and the first-order upwind discretisation satisfy the first and second properties. The remaining two schemes exhibit some form of dispersion.

Details about central finite difference and first-order upwind scheme are given in the Sect. D.





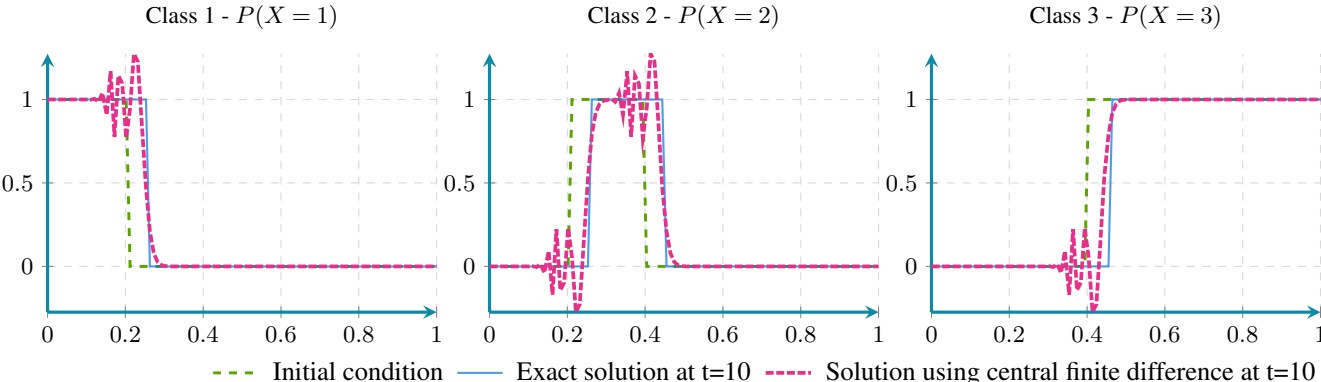

**Figure C1.** The advection of probabilities using central finite differences discretisation presents a dispersion effect

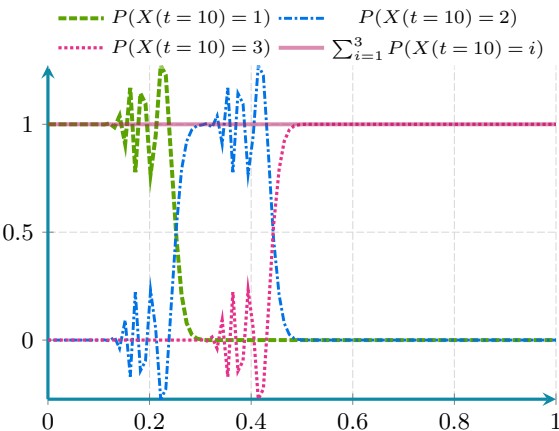

**Figure C2.** The probability conservation property is maintained even in presence of dispersion effects.

## Appendix D:  discretisation schemes

In this appendix section, we will derive the equivalent equation of central differences and upwind scheme applied to the
following advection equation:

$$\frac{\partial F(x,t)}{\partial t} + u\frac{\partial F(x,t)}{\partial x} = 0 \tag{D1}$$



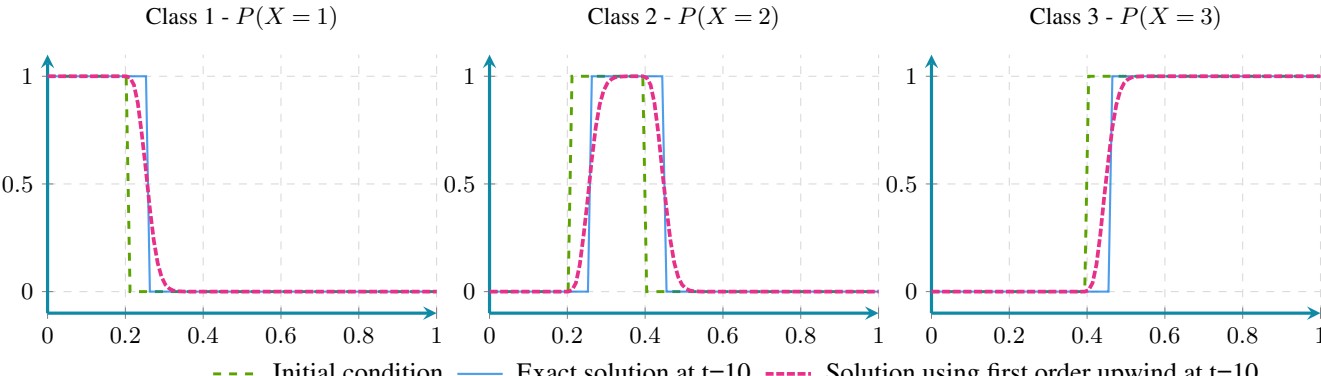

**Figure C3.** The advection of probabilities using first order upwind discretisation presents a diffusion effect

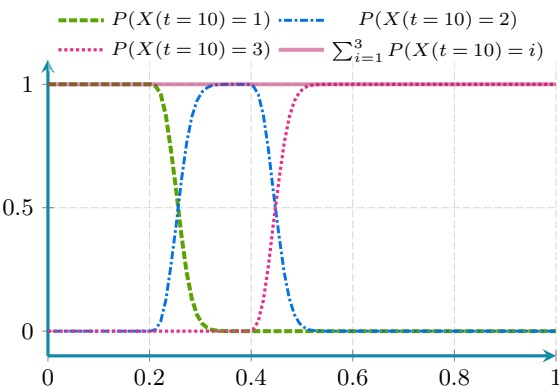

**Figure C4.** The probability conservation property is maintained even in presence of diffusion effects

## D1 Central Differences - Equivalent equation

We consider the second-order central discretisation in space and a first-order explicit forward difference in time applied to the advection equation.

$$\frac{F_i^{n+1} - F_i^n}{\Delta t} + u_i \frac{F_{i+1} - F_{i-1}}{2\Delta x} = 0 \tag{D2}$$

Using the Taylor formulas in the Eq. (D2), we get:

$$\partial_t F + \frac{\Delta t}{2} \partial_t^2 F + \mathcal{O}(\Delta t^2) = -u \left( \partial_x F - \frac{\Delta x^2}{6} \partial_x^3 F + \mathcal{O}(\Delta x^2) \right) \tag{D3}$$



Whereas we require just a first-order expansion in time, we can replace the second-order time derivative by another term coming from a Taylor first order expansion of the Eq. (D2) :

$$\partial_t(\partial_t F) + \mathcal{o}(\Delta t) = -\partial_t(u\partial_x F) + \mathcal{o}(\Delta x) \tag{D4}$$

Then,

$$\partial_t^2 F = -\partial_t u\partial_x F - u\partial_{xt}^2 F + \mathcal{o}(\Delta t, \Delta x)$$

Using the same approach, as in the Eq. (D4), the derivative $\partial_{xt}^2 F$ can be computed as follows :

$$\partial_x(\partial_t F) = -\partial_x u\partial_x F - u\partial_x^2 F + \mathcal{o}(\Delta t, \Delta x)$$

We replace the derivative $\partial_{xt}^2 F$ in the last formula :

$$\partial_t^2 F = -\partial_t u\partial_x F - u\left(-\partial_x u\partial_x F - u\partial_x^2 F\right) + \mathcal{o}(\Delta t, \Delta x) \tag{D5}$$

Finally, we replace the second-order derivative in the Eq. (D3) by the expression in Eq. (D5) :

$$\partial_t F + \frac{\Delta t}{2}\left(-\partial_t u\partial_x F - u\left(-\partial_x u\partial_x F - u\partial_x^2 F\right)\right)$$
$$= -u\left(\partial_x F - \frac{\Delta x^2}{6}\partial_x^3 F\right) + \mathcal{o}\left(\Delta t^2, \Delta x^2\right)$$

Hence,

$$\partial_t F + \tilde{u}\partial_x F = -\frac{\Delta t}{2}u^2\partial_x^2 F + \frac{\Delta x^2}{6}u\partial_x^3 F + \mathcal{o}\left(\Delta t^2, \Delta x^2\right), \tag{D6}$$

where $\tilde{u} = u - \frac{\Delta t}{2}\partial_t u + \frac{\Delta t}{2}u\partial_x u$.

## D2 First order upwind scheme - Equivalent equation

Now let's consider the first-order upwind discretisation of the spatial term, given by:

$$\begin{cases} \dfrac{F_i^{n+1} - F_i^n}{\Delta t} + u\dfrac{F_i - F_{i-1}}{\Delta x} = 0 & \text{if} \quad u \geq 0 \\ \dfrac{F_i^{n+1} - F_i^n}{\Delta t} + u\dfrac{F_{i+1} - F_i}{\Delta x} = 0 & \text{if} \quad u < 0 \end{cases}$$

These two equations can be written as:

$$\frac{F_i^{n+1} - F_i^n}{\Delta t} + \left(u_i^+\frac{F_i - F_{i-1}}{\Delta x} + u_i^-\frac{F_{i+1} - F_i}{\Delta x}\right) = 0, \tag{D7}$$

where $u_i^+ = \max(u_i, 0)$ and $u_i^- = \min(u_i, 0)$.

Considering the case of $u \geq 0$ of Eq. (D7), using the Taylor formulas, we get:

$$\partial_t F + \frac{\Delta t}{2}\partial_t^2 F + \mathcal{o}\left(\Delta t^2\right) = -u\left(\partial_x F - \frac{\Delta x}{2}\partial_x^2 F + \mathcal{o}\left(\Delta x^2\right)\right) \tag{D8}$$



As in the case of the central differences, we replace the second-order derivative $\partial_t^2 F$ in the Eq. (D8) by the expression in Eq. (D5). :

$$\partial_t F + \frac{\Delta t}{2} \left( -\partial_t u \partial_x F - u \left( -\partial_x u \partial_x F - u \partial_x^2 F \right) \right)$$
$$= -u \left( \partial_x F - \frac{\Delta x}{2} \partial_x^2 F \right) + \mathcal{o}\left( \Delta t^2, \Delta x^2 \right)$$

Hence,

$$\partial_t F + \tilde{u} \partial_x F = v_{num} \partial_x^2 F + \mathcal{o}\left( \Delta t^2, \Delta x^2 \right), \tag{D9}$$

where $\tilde{u} = u - \frac{\Delta t}{2} \partial_t u + \frac{\Delta t}{2} u \partial_x u$, and $v_{num} = \frac{u}{2} \left( \Delta x - u \Delta t \right)$ the introduced numerical viscosity/diffusion.

The equivalent equation of the second case of Eq. (D7) (case $u \leq 0$) is written as:

$$\partial_t F + \tilde{u} \partial_x F = v_{num} \partial_x^2 F + \mathcal{o}\left( \Delta t^2, \Delta x^2 \right), \tag{D10}$$

where $v_{num} = \frac{u}{2} \left( -\Delta x - u \Delta t \right)$

From Eq. (D9) and Eq. (D10) we can write the equivalent equation as:

$$\partial_t F + \tilde{u} \partial_x F = v_{num} \partial_x^2 F + \mathcal{o}\left( \Delta t^2, \Delta x^2 \right), \tag{D11}$$

where $\tilde{u} = u - \frac{\Delta t}{2} \partial_t u + \frac{\Delta t}{2} u \partial_x u$, and $v_{num} = \frac{u}{2} \left( sign(u) \Delta x - u \Delta t \right)$ the introduced numerical viscosity/diffusion.

**D3   Conclusion**

It should be noted that the finite central difference scheme exhibits instability due to the presence of negative diffusion in the second term in the Eq. (D6). However, by using a temporal scheme of higher order than 2, the negative diffusion term in $\Delta t$ can be eliminated, rendering the scheme stable. Nevertheless, the scheme becomes dispersive due to the third-order spatial derivative term, resulting in oscillations during the propagation of non-smooth signals, such as a front or Heaviside function.

Alternatively, the first-order upwind scheme offers stability but introduces numerical diffusion, affecting the accuracy of the solution, this diffusion is due to the second order derivative term in Eq. (D11).

Finally, the choice of the numerical scheme depends on the specific requirements of the problem, such as the desired accuracy and stability of the solution. To respect the properties described above, we use the first-order upwind scheme, as it doesn't introduce oscillations in the solution. The first-order upwind scheme is also easy to implement in a differentiable mode. Despite the limitation on the time step linked to the CFL condition, we consider it as a more appropriate scheme to integrate probability advection in a neural network.



*Author contributions.* R. El Montassir implemented the code, performed the experiments, and wrote the first draft of the manuscript. C. Lapeyre and O. Pannekoucke supervised the work. All authors collaborated on the design of the models and contributed to the manuscript's writing.

*Competing interests.* The authors declare having no competing interests.

*Acknowledgements.* We extend our gratitude to CERFACS for funding this work and providing computing resources, and to EUMETSAT and Météo France for providing essential data. Our sincere thanks to Luciano Drozda, Léa Berthomier and Bruno Pradel for constructive discussions and feedbacks, and to Isabelle d'Ast for technical support.



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
