# Peer review of "HyPhAICC v1.0: A Hybrid Physics-AI approach for probability fields advection, application to cloud cover nowcasting"

_EGUsphere, 2023_

## Referee Comment (RC1)

**REVIEW OF: HYPHAI V1.0: HYBRID PHYSICS-AI ARCHITECTURE FOR CLOUD COVER NOWCASTING**

In this paper, the authors introduced a novel hybrid model for cloud cover nowcasting. They utilize the advection equation to advance cloud cover maps, with velocities estimated through a neural network. The network parameters are optimized end-to-end to minimize cross-entropy loss between predicted cloud type probabilities and actual ones at subsequent time steps. I find the overall concept of the paper both innovative and appealing. Furthermore, the authors have included a range of results and diagnostics to evaluate the effectiveness of their methodology. Overall, I believe the paper is suitable for the GMD audience. However, there are certain aspects, particularly concerning the paper's writing style, that could benefit from further refinement.

Major comments :

1. Introduction and positioning of the paper: I appreciate the authors' introduction and I think that it effectively sets the stage for the study. However, I believe additional refinement could be beneficial. For example, around line 65, the authors discuss residual modeling as a methodology to mitigate imperfections in physics-based models. From my understanding, residual modeling in this context means using machine learning to correct errors of physical models. I suggest to avoid using the term "residual modeling" here, as it could potentially be confused with residual networks.

   In the same context, the authors mention that model correction "does not have the ability to enforce physics-based constraint." Correcting model errors can be approached in various ways. For instance, one approach involves defining machine learning-based parameterizations of physical models, where deep learning doesn't model a residual but instead represents missing physical processes. Recent advancements in the field have explored incorporating various physical constraints into such deep learning-based corrections. I suspect that this is what the authors are referring to in lines 70 to 80. I suggest that the authors revise this section and include references to recent state-of-the-art research to provide further context and clarity.

2. The majority of section two is not necessary and should be removed. It is quite standard to write numerical schemes in PyTorch. You can just mention in section three that you implemented your numerical advection scheme in PyTorch, and you can discuss the space and time discretizations you used in the appendix. Everything else including the discussion on automatic differentiation, gradient decent and the subsection 2.1 on combining neural nets with physics should be removed.

3. I also suggest to trim down the text in the remaining sections for example: around line 330 you should remove "which rely on computing the loss gradients with respect

to the model parameters. These gradients guide the update of the model's weights during the training process"

4. The authors developed 4 different versions of their model based on different parameterizations of the source term. However, in the experiments, only two configurations, the HyPhAI-1 and 2 are tested. If you plan to only test these two configurations, you should remove HyPhAI-3 and 4.

I hope that the authors will understand my comments in a constructive way, and that I value their work and the time they invested in the preparation of the manuscript. It might be that I have misunderstood something, in this case, if something wasn't clear for me as a reviewer, it is possible that it wouldn't be clear also for the readers.

---

## Referee Comment (RC2)

**HyPhAI v1.0: Hybrid Physics-AI architecture for cloud cover nowcasting – Review report**

**6th May 2024**

In this manuscript, the authors propose a new method that combines a neural network and a numerical integration scheme for cloud cover nowcasting. The innovative part of the method is that it is built on top of a cloud cover classification and hence relies on probabilistic approaches (in particular the network is trained using cross-entropy). The method is applied to a large dataset of real satellite images which have been labelled beforehand.

Overall I found the manuscript well written and easily understandable, with interesting methodological developments. I am therefore really positive about this manuscript.

**1 General comments**

**1.1 Objective of the manuscript**

After reading the entire manuscript, I am not entirely sure what is the objective. The title suggest that the objective is to develop and release a version of a geoscientific model, namely HyPhAI. On the other hand the last paragraphs of the introduction, and more generally the entire manuscript, leave the impression that the authors want to develop a method and illustrate it to a specific application (cloud cover nowcasting).

Finally, even though the name 'HyPhAI' corresponds to the model that is described here (which is indeed an hybrid model mixing physics and AI), this name could fit any hybrid model mixing physics and AI and hence it gives the false impression that the model describes all possible solutions for hybridising physics and AI. Therefore, I would suggest to be slightly more humble in choosing the name of the model.

**1.2 Background in machine learning and numerical schemes for PDEs**

A large part of section 2 is widely known in the community and, if needed by a reader, can be found in standard textbooks. In particular, I would suggest to remove

- in subsection 2.2 the last two paragraphs, from L 124 'During this training process...' to L 136 '... employed in neural networks.';

- the entire subsection 2.3;

- the entire subsection 2.4;

- the entire subsection 2.5;

or at the very least to put all these materials in appendices.

**1.3 Two or four model variants**

In subsection 3.2, four model variants are presented, whereas in the numerical experiments, only the first two are tested. Therefore, I would highly recommend to remove HyPhAI-3 and -4 from here to simplify the presentation (which is already rather complex). Then HyPhAI-3 and -4 could be either presented in appendices or in the discussion.

**1.4 Application to the Earth's full disk**

This inference experiment is very nice, but raises two key questions:

1. What is exactly a 'full disk' and how is it projected into the $3712 \times 3712$ squared image?

2. Beyond the visual impressions from figures 12 and A3, do you have scores to support the claim of 'remarkable adaptation' (L 481) and 'accurate and reliable' (L482)?

**2 Technical comments and suggestions**

**L 17-18** 'However, NWP models have inherent limitations in their ability to capture small-scale weather phenomena such as thunderstorms, tornadoes, and localised heavy rainfall events.' Please add a citation here.

**L 33** 'SHI et al., 2015' Is the capitalisation of the name intentional?

**L 37** 'This network excels' I would rather speak of 'neural architecture' as LSTM is no unique network.

**L 52** 'the hybridisation available techniques' → 'the available hybridisation techniques'?

**L 52-53** 'As discussed by Willard et al. (2022), the hybridisation available techniques leverage different aspects of ML models, e.g. the cost function, the design of the architecture or the weights' initialisation.' I would suggest to also cite here the review by Cheng et al. (2023).

**68-69** 'However, it does not have the ability to enforce physics-based constraints, as it primarily deals with errors rather than physical states.' Why not? Even in residual modelling, nothing prevents you from adding enforce a physics-based constraint.

**L 70** 'An advanced variation of residual modelling involves the integration of physics-based models and ML models.' I am not sure to see the difference here. In what you call residual modelling, the ML model predicts the errors of the physics-based model, in such a way that the final model is hybrid and aggregates the contribution of the physics-based model and of the ML model, which is precisely what you describe in the second part of this sentence.

**Equation 4** The notation $f_\theta(x_k)$ here is inconsistent with the notation $f_\theta(x, x_{\mathrm{Phy}})$ in Eq. 3.

**L 122-123** 'The choice of $l$ depends, among other things, on the statistical model $f_\theta$.' Here I disagree. The choice of the likelihood should not depend on the model, but it should be the other way around: the choice of the model should be made in order to be able to minimise the likelihood.

**Figures 3 and 4** Is this a game of 'find 7 differences'? On a more serious note, I wonder whether these two figures could be merged.

**L 234** 'These $256 \times 256$ satellite images' I assume that $256 \times 256$ is the size of each image, but here one could naively understand that there are in total 65536 images.

**L 260** 'We have demonstrated in Appendix C' Using the past tense feels a bit weird here. I would recommend using the present.

**L 267** 'to check the Courant-Friedrichs-Lewy (CFL) condition' $\rightarrow$ 'to satistfy the Courant-Friedrichs-Lewy (CFL) condition'

**L 270** 'It takes previous observations'. How many observations in the past? How do you merge the information from all these observations? Are they stacked in the channel direction?

**L 273** 'doesn't' $\rightarrow$ 'does not'.

**L 316-317** 'with a total of approximately 100,000 images.' I think that it would be better and in that case even shorter to give the exact number.

**L 320** 'After cleaning' Please describe this cleaning step. Furthermore, please also describe what rule you use to split between training and validation.

**L 404-405** 'In the comparative evaluation, we included the widely used U-Net' If I understood correctly, you used as baseline a 'vanilla' U-Net. Why not using a U-Net Xception style as for HyPhAI?

**Figure 7**   While I agree that confidence intervals are in general needed, here they make the figure unreadable: there is just too much information. Furthermore, what confidence intervals are these: 99%? 95%? 90%? other?

**Table 1**   Please describe what bold font means.

**Figure 8**   Labels are too small on this figure. What class correspond to colour 'beige' (which can be seen e.g. North-East of France)? Finally, the projection for this map seems a bit weird (possibly flattened in the latitude direction).

**L 440-441**   'highlighting that HyPhAI-1 produces more realistic and less blurry forecasts compared to the U-Net' Rigorously speaking this statement is true, but in my opinion it is a bit misleading because it hides the fact that even with HyPhAI-1 the prediction are much smoother than the truth.

**L 445-446**   'the lost details in HyPhAI-1's predictions are only due to the diffusion added numerically by the discretisation scheme used' Are you sure about this statement? As far as I know, many ML models trained with the point-wise metrics tend to yield smooth predictions because of the double penalty issue (see, e.g., Bonavita 2023). I suspect that this is the case in your model. If not, and hence if numerical diffusion is the only obstacle, then can't you use another numerical scheme with less numerical diffusion?

**Figure 9**   Why did you use a different extent for this map?

**Figure 10**   I think that figure 10 is discussed after figure 11 in the text.

**Section 4.4**   Can we really draw robust conclusions here by looking at the validation scores (and not the test scores)? Furthermore, the fact that hybrid models are usually more accurate with less training data is already widely known in the hybrid modelling literature (see, again, Cheng et al. 2023).

**L 476**   'earth' → 'Earth'.

**L 478**   'This expansive full disk domain is 14 times the size of the training area.' If I am not mistaken, the original domain is $256 \times 256$ and the full disk is $3712 \times 3712$ (BTW, this information is only mentioned in the caption of Fig. A3, it would be better to mention it in the text). The full disk is therefore 210.25 times bigger, right?

**Figure 12**   This figure is not referenced in the text.

**References**

Bonavita, Massimo (2023). 'On Some Limitations of Data-Driven Weather Forecasting Models'. Version 2. In: DOI: 10.48550/arxiv.2309.08473.

Cheng, Sibo et al. (June 2023). 'Machine Learning With Data Assimilation and Uncertainty Quantification for Dynamical Systems: A Review'. In: *IEEE/CAA Journal of Automatica Sinica* 10.6, pp. 1361–1387. DOI: 10.1109/JAS.2023.123537.

---

## Author Comment (AC1)

**Response to reviewer**

R. El Montassir, O. Pannekoucke and C. Lapeyre

June 4, 2024

We thank the anonymous reviewer for his/her feedback on our manuscript. We appreciate the positive remarks and constructive suggestions. We are grateful for the time and effort that he/she invested in evaluating our work.

We copied the reviewer's commentary below, and we replied in teal to each point. We also provide the changes made in the manuscript for each comment.

1. *Introduction and positioning of the paper: I appreciate the authors' introduction and I think that it effectively sets the stage for the study. However, I believe additional refinement could be beneficial. For example, around line 65, the authors discuss residual modeling as a methodology to mitigate imperfections in physics-based models. From my understanding, residual modeling in this context means using machine learning to correct errors of physical models. I suggest to avoid using the term "residual modeling" here, as it could potentially be confused with residual networks. In the same context, the authors mention that model correction "does not have the ability to enforce physics-based constraint." Correcting model errors can be approached in various ways. For instance, one approach involves defining machine learning-based parameterizations of physical models, where deep learning doesn't model a residual but instead represents missing physical processes. Recent advancements in the field have explored incorporating various physical constraints into such deep learning-based corrections. I suspect that this is what the authors are referring to in lines 70 to 80. I suggest that the authors revise this section and include references to recent state-of-the-art research to provide further context and clarity.*

   - Even if the term is explained in the text, we agree that the term "residual modelling" can be misleading, we have removed it. For the second point, adding data driven parametrisations is not part of what we called residual modelling, but, as you mentioned, it is part of the next category of methods that is more general and includes residual modelling. Therefore, the statement "does not have the ability to enforce physics-based constraint" is still accurate. However, this sentence is misleading and not necessary; we removed it. A figure (Fig. 1) has been added to clarify this concept of residual modelling.

   "To address imperfections in physics-based models, a common strategy is  error modelling. Here, an ML model learns to predict the errors (also called residuals) made by the physics-based model (Forssell and Lindskog, 1997). This approach leverages learned biases to correct predictions (see Fig. 1.).

    A more general approach that does not deal only with errors is to create hybrid models merging physics-based models and ML models.  For example, in scenarios where the dynamics of  physics is fully defined,  the output of a physics-based model can be used as input to an ML model."

2. *The majority of section two is not necessary and should be removed. It is quite standard to write numerical schemes in PyTorch. You can just mention in section three that you implemented your numerical advection scheme in PyTorch, and you can discuss the space and time discretizations you used in the appendix. Everything else including the discussion on automatic differentiation, gradient decent and the subsection 2.1 on combining neural nets with physics should be removed.*

   - Similar comments were made by the other reviewer. We are aware that the section contains standard information, however, we do not fully agree that these details are known to the majority

of the readers, especially in our restricted community, meteorology. We believe that it is important to provide a clear and detailed explanation of the challenges encountered when combining neural networks with physics-based models, in order to provide a smooth reading experience for the reader. We have, however, moved the second section to the appendix and updated the introduction of the section 3 as follows:

"In this  work, we address applications involving dynamics with unknown variables that require estimation. For example, the cloud motion field is one of the unknown variables in the application considered. In such cases, as discussed in the introduction, a joint resolution approach is more appropriate. Here, the physical model uses the neural network outputs to compute predictions, integrating the two models as follows:

$$y = \phi \circ f_\theta \left( x \right),$$

where $x$ is the input, $f_\theta$ represents the neural network, $\phi$ denotes the physical model, and $y$ is the output. In this setup, $\phi$ implicitly imposes a hard constraint on the outputs, potentially accelerating the convergence of the neural network during training.

This method raises some trainability challenges as the physics-based model is involved in the training process, and it should be differentiable, in the sense of automatic differentiation, in order to allow the back-propagation of gradients (refer to Appendix B). We show in Appendix B how spatial derivatives of PDEs can be approximated within a neural network in a differentiable way using convolution operations. This allows us to compute gradients and back-propagate them during the training process. This fundamental knowledge serves as a foundation for our investigation of novel hybrid Physics-AI architectures. With these established principles, we present in this section the proposed hybrid architecture, which is applied to cloud cover nowcasting. In this section, we introduce our hybrid Physics-AI architecture,  detailed in Sect.  2.1. Section 2.2 explains the different physical modelling approaches investigated in this study. Following that, Sect.  2.3, Sect.  2.4 and Sect.  2.5 sequentially present the training procedure, evaluation metrics, and benchmarking procedure."

3. *I also suggest to trim down the text in the remaining sections for example: around line 330 you should remove "which rely on computing the loss gradients with respect to the model parameters. These gradients guide the update of the model's weights during the training process".*

   - Done.

   "The training of the model parameters is achieved through gradient-based methods. Here, Adam optimiser [Kingma and Ba, 2017] is used with a learning rate of $10^{-3}$ and a batch size of 4 with 16 accumulation steps,  allowing us to simulate a batch size of 64. The training was performed using a single Nvidia A100 GPU for 30 epochs."

4. *The authors developed 4 different versions of their model based on different parameterizations of the source term. However, in the experiments, only two configurations, the HyPhAI-1 and 2 are tested. If you plan to only test these two configurations, you should remove HyPhAI-3 and 4.*

   - The same point was raised by the other reviewer. The reason why these two versions were presented even if they didn't show any improvement is to show the flexibility of the model. We added a sentence to clarify this and we moved these two versions to the appendix.

   "The second version of the hybrid model, denoted  HYPHAICCAST-2, adds this source term to the advection. This modelling is described in the following equations:

$$\partial_t P_j + \overrightarrow{V} \cdot \overrightarrow{\nabla} P_j = \tanh(S_j) \quad \forall j \in \{1, 2, \ldots, C\}, \tag{1}$$

where $S_j$ is estimated using a second U-Net model (see Fig. 4). While the previous modelling describes the missing physical process in the advection, it does not satisfy the probability conservation property. Thus, this modelling does not conserve the probabilistic nature of P over time. To ensure the appropriate dynamics of probability, a robust framework is provided by continuous-time

Markov processes across finite states [Pavliotis and Stuart, 2008, chap. 5]. In this framework, the probability trend is controlled by a linear dynamics, ensuring the bound preservation, positivity, and probability conservation. Two other models based on this framework, named HyPhAICCast-3 and HyPhAICCast-4, are presented in the Appendix A1 and Appendix A2. However, these models did not show any performance improvement compared to the simpler HyPhAICCast-1. Indeed, beyond the performance aspect, this hybridisation framework is flexible, not only limited to the advection, and can be extended to other physical processes"

**References**

[Kingma and Ba, 2017] Kingma, D. P. and Ba, J. (2017). Adam: A Method for Stochastic Optimization. arXiv:1412.6980 [cs].

[Pavliotis and Stuart, 2008] Pavliotis, G. and Stuart, A. (2008). *Multiscale Methods: Averaging and Homogenization*, volume 53. Springer, New York, NY.

---

## Author Comment (AC2)

**Response to reviewer**

R. El Montassir, O. Pannekoucke and C. Lapeyre

June 4, 2024

We thank the reviewer, Dr. Alban Farchi, for his feedback on our manuscript. We appreciate the positive remarks and constructive suggestions. We are grateful for the time and effort that he invested in evaluating our work.

We copied the reviewer's commentary below, and we replied in teal to each point. We also provide the changes made in the manuscript for each comment.

**1 General comments**

**1.1 Objective of the manuscript**

*After reading the entire manuscript, I am not entirely sure what is the objective. The title suggest that the objective is to develop and release a version of a geoscientific model, namely HyPhAI. On the other hand the last paragraphs of the introduction, and more generally the entire manuscript, leave the impression that the authors want to develop a method and illustrate it to a specific application (cloud cover nowcasting).*

*Finally, even though the name 'HyPhAI' corresponds to the model that is described here (which is indeed an hybrid model mixing physics and AI), this name could fit any hybrid model mixing physics and AI and hence it gives the false impression that the model describes all possible solutions for hybridising physics and AI. Therefore, I would suggest to be slightly more humble in choosing the name of the model.*

- Regarding the objective of this work, it is to develop a hybrid architecture that combines physics and AI that can be applied to a wide range of geoscientific problems. The cloud cover nowcasting is just an application of this architecture. We agree that the title was misleading, and we changed it to "A Hybrid Physics-AI (HyPhAI) approach for probability fields advection: Application to cloud cover nowcasting". We also changed the names of the models to HyPhAICCast-1, HyPhAICCast-2, HyPhAICCast-3, and HyPhAICCast-4.

**1.2 Background in machine learning and numerical schemes for PDEs**

*A large part of section 2 is widely known in the community and, if needed by a reader, can be found in standard textbooks. In particular, I would suggest to remove*

- *in subsection 2.2 the last two paragraphs, from L 124 'During this training process...' to L 136 '... employed in neural networks.';*

- *the entire subsection 2.3;*

- *the entire subsection 2.4;*

- *the entire subsection 2.5;*

*or at the very least to put all these materials in appendices.*

- Similar comments were made by the other reviewer. We are aware that the section contains standard information; however, we do not fully agree that these details are known to the majority of the readers, especially in our restricted community, meteorology. We believe that it is important to provide a clear and detailed explanation of the challenges encountered when combining neural networks with physics-based models, in order to provide a smooth reading experience for the reader. However, we have moved this Section 2 to the Appendix and updated the introduction of Section 3 as follows:

"In this  work, we address applications involving dynamics with unknown variables that require estimation. For example, the cloud motion field is one of the unknown

variables in the application considered. In such cases, as discussed in the Introduction, a joint resolution approach is more appropriate. Here, the physical model uses the neural network outputs to compute predictions, integrating the two models as follows:

$$y = \phi \circ f_\theta\left(x\right),$$

where $x$ is the input, $f_\theta$ represents the neural network, $\phi$ denotes the physical model, and $y$ is the output. In this setup, $\phi$ implicitly imposes a hard constraint on the outputs, potentially accelerating the convergence of the neural network during training.

This method raises some trainability challenges as the physics-based model is involved in the training process, and it should be differentiable, in the sense of automatic differentiation, in order to allow the back-propagation of gradients (refer to Appendix B). We show in Appendix B how spatial derivatives of PDEs can be approximated within a neural network in a differentiable way using convolution operations. This allows us to compute gradients and back-propagate them during the training process. This fundamental knowledge serves as a foundation for our investigation of novel hybrid Physics-AI architectures. With these established principles, we present in this section the proposed hybrid architecture, which is applied to cloud cover nowcasting. In this section, we introduce our hybrid Physics-AI architecture,  detailed in Sect.  2.1. Section 2.2 explains the different physical modelling approaches investigated in this study. Following that, Sect. 2.3, Sect. 2.4 and Sect. 2.5 sequentially present the training procedure, evaluation metrics, and benchmarking procedure."

**1.3 Two or four model variants**

*In subsection 3.2, four model variants are presented, whereas in the numerical experiments, only the first two are tested. Therefore, I would highly recommend to remove HyPhAI-3 and -4 from here to simplify the presentation (which is already rather complex). Then HyPhAI-3 and -4 could be either presented in appendices or in the discussion.*

- The same point was raised by the other reviewer. The reason why these two versions were presented even if they didn't show any improvement is to show the flexibility of the model. We added a sentence to clarify this, and we moved these two versions to the appendix.

"The second version of the hybrid model, denoted HYPHAICCAST-2, adds this source term to the advection. This modelling is described in the following equations:

$$\partial_t P_j + \overrightarrow{V} \cdot \overrightarrow{\nabla} P_j = \tanh(S_j) \quad \forall j \in \{1, 2, \ldots, C\}, \tag{1}$$

where $S_j$ is estimated using a second U-Net model (see Fig. 4). While the previous modelling describes the missing physical process in the advection, it does not satisfy the probability conservation property. Thus, this modelling does not conserve the probabilistic nature of P over time. To ensure the appropriate dynamics of probability, a robust framework is provided by continuous-time Markov processes across finite states [Pavliotis and Stuart, 2008, chap. 5]. In this framework, the probability trend is controlled by a linear dynamics, ensuring the bound preservation, positivity, and probability conservation. Two other models based on this framework, named HyPhAICCast-3 and HyPhAICCast-4, are presented in the Appendix A1 and Appendix A2. However, these models did not show any performance improvement compared to the simpler HyPhAICCast-1. Indeed, beyond the performance aspect, this hybridisation framework is flexible, not only limited to the advection, and can be extended to other physical processes"

**1.4 Application to the Earth's full disk**

*This inference experiment is very nice, but raises two key questions:*

1. *What is exactly a 'full disk' and how is it projected into the $3712 \times 3712$ squared image?*

   - A satellite full disk view of the Earth is a single image taken by a satellite in geostationary orbit, capturing an entire hemisphere of the Earth's surface in one frame. The $3712 \times 3712$ image simply corresponds to the image captured by the satellite, in this case the satellite is a geostationary satellite called Meteosat Second Generation (MSG) provided by EUMETSAT and positioned at 0° longitude. Here is the modified sentence:

"..., we tested it on a much larger domain,  an entire hemisphere of the Earth - also called a full disk - centred at 0 degrees longitude.  The satellite observations of this expansive full disk domain  are of size $3712 \times 3712$, which is 210.25 times larger than the size of the training ones."

2. *Beyond the visual impressions from figures 12 and A3, do you have scores to support the claim of 'remarkable adaptation' (L 481) and 'accurate and reliable' (L482)?*

   - We agree that the visual impressions are not sufficient to support these claims, however it's irrelevant to compare scores over different domains. We rephrased the sentences:

   "..., we tested it on a much larger domain,  an entire hemisphere of the Earth - also called a full disk - centred at 0 degrees longitude.  The satellite observations of this expansive full-disk domain are of size $3712 \times 3712$, which is 210.25 times larger than the size of the training ones. It has diverse meteorological conditions and includes projection deformations when mapped onto a two-dimensional plane., while the extreme deformations at the edge of the disk make this data less useful for operation purposes, it still provides an interesting testing ground for HyPhAICCast-1's generalisation ability. In this analysis, we focus only on visual aspects. Despite the significant differences between the training domain and the full disk, we observed  good qualitative forecasts of the HyPhAICCast-1 model on this new domain without any specific training on it (see Fig. 12 and Fig. A4). The cloud motion estimation on the full disk was found to be visually consistent, a video supplement is provided in the supplementary material. This successful transferability of the model highlights its potential robustness ..."

**2 Technical comments and suggestions**

**L 17-18** *'However, NWP models have inherent limitations in their ability to capture small-scale weather phenomena such as thunderstorms, tornadoes, and localised heavy rainfall events.' Please add a citation here.*

   - We added the following references: [Schultz et al., 2021, Matte et al., 2022, Joe et al., 2022].

**L 33** *'SHI et al., 2015' Is the capitalisation of the name intentional?*

   - Corrected.

**L 37** *'This network excels' I would rather speak of 'neural architecture' as LSTM is no unique network.*

   - Corrected.

**L 52** *'the hybridisation available techniques' → 'the available hybridisation techniques'?*

   - Done.

**L 52-53** *'As discussed by Willard et al. (2022), the hybridisation available techniques leverage different aspects of ML models, e.g. the cost function, the design of the architecture or the weights' initialisation.' I would suggest to also cite here the review by [Cheng et al., 2023].*

   - Done.

**L 68-69** *'However, it does not have the ability to enforce physics-based constraints, as it primarily deals with errors rather than physical states.' Why not? Even in residual modelling, nothing prevents you from adding enforce a physics-based constraint.*

   - Yes, but it would not be in this category of methods, either in the first one (L 55-60) or in the next one. However, this sentence is misleading and not necessary, we removed it.

**L 70**   'An advanced variation of residual modelling involves the integration of physics based models and ML models.' I am not sure to see the difference here. In what you call residual modelling, the ML model predicts the errors of the physics-based model, in such a way that the final model is hybrid and aggregates the contribution of the physics-based model and of the ML model, which is precisely what you describe in the second part of this sentence.

- Residual modelling is a specific case, and what we describe in the following lines is more general. We rephrased the sentence to make it clearer:

"To address imperfections in physics-based models, a common strategy is  error modelling. Here, an ML model learns to predict the errors (also called residuals) made by the physics-based model (Forssell and Lindskog, 1997). This approach leverages learned biases to correct predictions (see Fig. 1.).

 A more general approach that does not deal only with errors is to create hybrid models merging physics-based models and ML models.  For example, in scenarios where the dynamics of  physics is fully defined,  the output of a physics-based model can be used as an input to an ML model."

**Equation 4**   The notation $f_\theta(x_k)$ here is inconsistent with the notation $f_\theta(x, x_{Phy})$ in Eq. 3 .

- We thank the reviewer for pointing this out, we replaced $\phi \circ f_\theta(x_k)$ by $f_\theta(x_k)$.

**L 122-123**   'The choice of $l$ depends, among other things, on the statistical model $f_\theta$.' Here I disagree. The choice of the likelihood should not depend on the model, but it should be the other way around: the choice of the model should be made in order to be able to minimise the likelihood.

- We thank the reviewer for pointing this confusing sentence out. What we meant is for example, in image generation, the MSE loss can be used for a DDPM model, while GAN require a completely different loss function. But we agree that this could be seen in the other way around, and that "statistical" is just adding confusion, we removed this sentence.

**Figures 3 and 4**   Is this a game of 'find 7 differences'? On a more serious note, I wonder whether these two figures could be merged.   - We don't see how these two figures could be merged, each one shows a different model. The difference between the two figures is that Figure 4 adds a source term to the equation, and this source term is estimated by a U-Net, this is explained in the caption and also in the text (subsection 3.2.2). However, we have reduced the opacity of the unchanged parts in the second diagram and highlighted the additional parts.

**L 234**   'These $256 \times 256$ satellite images' I assume that $256 \times 256$ is the size of each image, but here one could naively understand that there are 65536 images in total.

- We rephrased the sentence:

"..., the time step is 15 minutes  and each image is of size $256 \times 256$. These images have been processed..."

**L 260**   'We have demonstrated in Appendix C' Using the past tense feels a bit weird here. I would recommend using the present.

- Done.

**L 267**   'to check the Courant-Friedrichs-Lewy (CFL) condition' → 'to satistfy the Courant-Friedrichs-Lewy (CFL) condition'

- Done.

**L 270**   'It takes previous observations'. How many observations in the past? How do you merge the information from all these observations? Are they stacked in the channel direction?

- We use the last 4 images, they are stacked in the channel direction. We added this information in the text:

"It takes  the last four observations stacked on the channel axis, and estimates ..."

**L 273**   *'doesn't' → 'does not'.*
    - Corrected here and in two other places.

**L 316-317**   *'with a total of approximately 100,000 images.' I think that it would be better and in that case even shorter to give the exact number.*
    - The exact number is 105,120, we added this information in the text:
    "The training was carried out on a dataset containing about three years of data from 2017 to 2019, with a total of  105 120 images."

**L 320**   *'After cleaning' Please describe this cleaning step. Furthermore, please also describe what rule you use to split between training and validation.*
    - The cleaning was done by removing the images with zero cloud cover and gathered all the sequences with 12 constructive images. The split between training and validation was done randomly. We added this information in the text and moved the sentence "To improve ... patterns." to the end of the paragraph, and divided it into two sentences:
    "The training was carried out on a dataset containing about three years of data from 2017 to 2019, with a total of  105 120 images. The images with zero cloud cover were removed, then we assembled all the sequences with 12 consecutive images. After this cleaning step, we randomly split the dataset, 8 224 sequences were used for training, and 432 for validation. The test set was performed on a separate dataset from the same region but from 2021.
    To improve the diversity of the training set and take into account a possible overfitting on the typical movements of clouds in the Western Europe region, we randomly applied simple transformations to the images, more precisely, rotations of 90, 180 and 270 degrees, which increased the dataset size and improved the model's ability to learn various cloud motion patterns."

**L 404-405**   *'In the comparative evaluation, we included the widely used U-Net' If I understood correctly, you used as baseline a 'vanilla' U-Net. Why not using a U-Net Xception style as for HyPhAI?*
    - We used the classical U-Net as a baseline because it is the one that is used is other works for the same task. We added this information in the text:
    "In the comparative evaluation, we included the  well-known U-Net [Ronneberger et al., 2015]. This classical U-Net is different from the one used to estimate the velocity in the proposed hybrid models (refer to Fig. 3 and Fig. 4). The choice of this classical U-Net for comparison is justified by the fact that it is the most widely used in the literature for the same task (e.g. [Ayzel et al., 2020, Berthomier et al., 2020, Trebing et al., 2021])."

**Figure 7**   *While I agree that confidence intervals are in general needed, here they make the figure unreadable: there is just too much information. Furthermore, what confidence intervals are these: 99%?95%?90% ? other?*
    - We believe that the confidence intervals are important to show the uncertainty in the scores. However, we understand that the figure is too crowded, thus we removed them and provided another figure with the confidence intervals in the Appendix. The treshold used for the confidence intervals is 99%. We added this information in the caption:
    "The confidence intervals were estimated using Bootstrapping, with a threshold of 99%."

**Table 1**   *Please describe what bold font means.*
    - The bold font indicates the best score. We added this information in the caption:
    "... (↑: higher is better, ↓: lower is better). The best scores are indicated in bold font.

**Figure 8**   *Labels are too small on this figure. What class correspond to colour 'beige' (which can be seen e.g. North-East of France)? Finally, the projection for this map seems a bit weird (possibly flattened in the latitude direction).*
    - We increased the size of the labels. The beige colour is not in the labels, it corresponds to the land areas. This information is added to the figure's caption. The projection used here is the plate carrée, hence the effect noticed in the reviewer's comment.

**L 440-441** *'highlighting that HyPhAI-1 produces more realistic and less blurry forecasts compared to the U-Net' Rigorously speaking this statement is true, but in my opinion it is a bit misleading because it hides the fact that even with HyPhAI-1 the prediction are much smoother than the truth.*

- We agree that point that the HyPhAICCast-1 predictions are smoother than the ground truth, and this is admitted in the conclusion. But here we are comparing the HyPhAICCast-1 predictions with the U-Net predictions, and we explained in the same paragraph the reason behind the HyPhAICCast-1 loss of details:

" The lost details in HyPhAICCast-1's predictions are only due to the  numerical scheme, in ideal conditions, the HyPhAICCast-1 should preserve the same details during the advection process, and there is no other trainable part in between that can smooth the predictions; however, the upwind discretisation used scheme adds a numerical diffusion and crushing the small cloud cells (refer to  Appendix E for more details). "

**L 445-446** *'the lost details in HyPhAI-1's predictions are only due to the diffusion added numerically by the discretisation scheme used' Are you sure about this statement? As far as I know, many ML models trained with the point-wise metrics tend to yield smooth predictions because of the double penalty issue (see, e.g., Bonavita 2023). I suspect that this is the case in your model. If not, and hence if numerical diffusion is the only obstacle, then can't you use another numerical scheme with less numerical diffusion?*

- We understand the suspicion of the reviewer. However, the only trainable component of HyPhAICCast-1 is the one used to estimate the velocity field, and this component does not have direct access to the loss function. The velocity field is used to advect the cloud cover field. In ideal conditions, we should preserve the same details during the advection process, and there is no other trainable part in between that can smooth the predictions. This information is added to the same paragraph.

" The lost details in HyPhAICCast-1's predictions are only due to the  numerical scheme, in ideal conditions, the HyPhAICCast-1 should preserve the same details during the advection process, and there is no other trainable part in between that can smooth the predictions; however, the upwind discretisation used scheme adds a numerical diffusion and crushing the small cloud cells (refer to  Appendix E for more details). "

- Regarding the use of another numerical scheme, it is worth noting that the scheme to use should be automatically differentiable, stable, and preserves both details and the probabilistic properties. At the moment, we do not have an alternative scheme that satisfies all these conditions. For example, the central differences preserve more details than the upwind scheme, even if we ignore the dispersion issues, but it is not stable for the last lead times as the dispersion issues become more important.

**Figure 9** *Why did you use a different extent for this map?*

- As the figure shows one image, we have more space to show it with a larger extent, and the choice of the orthographic projection here is more aesthetic.

**Figure 10** *I think that figure 10 is discussed after figure 11 in the text.*

- Corrected.

**Section 4.4** *Can we really draw robust conclusions here by looking at the validation scores (and not the test scores)? Furthermore, the fact that hybrid models are usually more accurate with less training data is already widely known in the hybrid modelling literature (see, again, [Cheng et al., 2023].*

- We understand the concerns of the reviewer. However, validation data are generally used to tune hyperparameters, and we consider the required data size to be one of these hyperparameters. And we presented the results of these experiments, we believe that we can draw robust conclusions based on them as these data are not seen by the model during training. Regarding the second point, we agree that it is a well-known fact that hybrid models can be data-efficient [Schweidtmann et al., 2024, Cheng et al., 2023], and we are not claiming this. We clarified this point in the text:

"This finding indicates that this hybrid model is remarkably data efficient, capable of delivering satisfactory performance even with limited training data, which has been highlighted by other studies [Schweidtmann et al., 2024, Cheng et al., 2023]. This quality is very important, particularly for tasks with insufficient provided data."

- However, we believe that it is important to show that in our context, and we provided a quantitative mesure of this efficiency.

**L 476** *'earth' → 'Earth'.*

    - Corrected here and in one other place.

**L 478** *'This expansive full disk domain is 14 times the size of the training area.' If I am not mistaken, the original domain is $256\times256$ and the full disk is $3712\times3712$ (BTW, this information is only mentioned in the caption of Fig. A3, it would be better to mention it in the text). The full disk is therefore 210.25 times bigger, right?*

    - Yes, it is 210.25 times larger (14 times in each direction). We changed this information in the text. The size of the full disk is now mentioned in the text:

    " The satellite observations of this expansive full-disk domain are of size $3712 \times 3712$, which is 210.25 times larger than the size of the training ones."

**Figure 12** *This figure is not referenced in the text.*

    - We added a reference to this figure in the text:

    Despite the significant differences between the training domain and the full disk, we observed  good qualitative forecasts of the HyPhAICCast-1 model on this new domain without any specific training on it (see Fig. 12 and Fig. A4).

**References**

[Ayzel et al., 2020] Ayzel, G., Scheffer, T., and Heistermann, M. (2020). RainNet v1.0: a convolutional neural network for radar-based precipitation nowcasting. *Geoscientific Model Development*, 13(6):2631–2644. Publisher: Copernicus GmbH.

[Berthomier et al., 2020] Berthomier, L., Pradel, B., and Perez, L. (2020). Cloud Cover Nowcasting with Deep Learning. In *2020 Tenth International Conference on Image Processing Theory, Tools and Applications (IPTA)*, pages 1–6. arXiv:2009.11577 [cs].

[Cheng et al., 2023] Cheng, S., Quilodrán-Casas, C., Ouala, S., Farchi, A., Liu, C., Tandeo, P., Fablet, R., Lucor, D., Iooss, B., Brajard, J., Xiao, D., Janjic, T., Ding, W., Guo, Y., Carrassi, A., Bocquet, M., and Arcucci, R. (2023). Machine Learning With Data Assimilation and Uncertainty Quantification for Dynamical Systems: A Review. *IEEE/CAA Journal of Automatica Sinica*, 10(6):1361–1387. Publisher: IEEE/CAA Journal of Automatica Sinica.

[Joe et al., 2022] Joe, P., Sun, J., Yussouf, N., Goodman, S., Riemer, M., Gouda, K. C., Golding, B., Rogers, R., Isaac, G., Wilson, J., Li, P. W. P., Wulfmeyer, V., Elmore, K., Onvlee, J., Chong, P., and Ladue, J. (2022). Predicting the Weather: A Partnership of Observation Scientists and Forecasters. In Golding, B., editor, *Towards the "Perfect" Weather Warning: Bridging Disciplinary Gaps through Partnership and Communication*, pages 201–254. Springer International Publishing, Cham.

[Matte et al., 2022] Matte, D., Christensen, J. H., Feddersen, H., Vedel, H., Nielsen, N. W., Pedersen, R. A., and Zeitzen, R. M. K. (2022). On the Potentials and Limitations of Attributing a Small-Scale Climate Event. *Geophysical Research Letters*, 49(16):e2022GL099481. _eprint: https://onlinelibrary.wiley.com/doi/pdf/10.1029/2022GL099481.

[Pavliotis and Stuart, 2008] Pavliotis, G. and Stuart, A. (2008). *Multiscale Methods: Averaging and Homogenization*, volume 53. Springer, New York, NY.

[Ronneberger et al., 2015] Ronneberger, O., Fischer, P., and Brox, T. (2015). U-Net: Convolutional Networks for Biomedical Image Segmentation. In Navab, N., Hornegger, J., Wells, W. M., and Frangi, A. F., editors, *Medical Image Computing and Computer-Assisted Intervention – MICCAI 2015*, volume 9351, pages 234–241. Springer International Publishing, Cham. Series Title: Lecture Notes in Computer Science.

[Schultz et al., 2021] Schultz, M. G., Betancourt, C., Gong, B., Kleinert, F., Langguth, M., Leufen, L. H., Mozaffari, A., and Stadtler, S. (2021). Can deep learning beat numerical weather prediction? *Philosophical transactions. Series A, Mathematical, physical, and engineering sciences*, 379(2194):20200097.

[Schweidtmann et al., 2024] Schweidtmann, A. M., Zhang, D., and von Stosch, M. (2024). A review and perspective on hybrid modeling methodologies. *Digital Chemical Engineering*, 10:100136.

[Trebing et al., 2021] Trebing, K., Stanczyk, T., and Mehrkanoon, S. (2021). SmaAt-UNet: Precipitation nowcasting using a small attention-UNet architecture. *Pattern Recognition Letters*, 145:178–186.

---

## Author Response (AR2)

**Author's response**

R. El Montassir, O. Pannekoucke and C. Lapeyre

July 4, 2024

Dear M. O'Brien,

We thank you for accepting our manuscript for publication in GMD. We are grateful for your constructive comments and suggestions, which have helped us improve the manuscript.

We have carefully read the comments and suggestions and have made the following changes to the manuscript.

**1 Major changes:**

- L. 284: Added more details about the EXIM product.

- L. 325: Added a reference to the EXIM documentation.

- Fig. 1: Updated the caption using the term "error modelling"

**Track-changes:** To facilitate the comparison between the two versions of the manuscript, a track-changes file lists all changes, where the old statements are coloured red and the new ones are coloured blue.

**Response to editor's comments**

We copied the editor's commentary below, we replied in teal to each point. We also provide the changes made in the manuscript for each comment.

1. *Reviewer 1 suggested avoiding use of the term "residual modeling" and I agree with their reasoning. Your revised manuscript avoids the use of this phrase in one place, but it remains in others (e.g., Figure 1). I suggest doing a global search-and-replace, using the phrase "error modeling" instead.*

   - Done.

2. *line 319 - "and EXIM is advecting the last observation while keeping the same level of details" ¡– I agree that's true, but HyPhAICC is also advecting the last observation, so this explanation seems incomplete. Is it because EXIM uses a more accurate (and less diffusive) advection scheme? Can you reference EXIM documentation that would support this argument? If not, the discussion here should be revised.*

   - Thank you for pointing this out. We have revised the section 2.5 to include more details about the EXIM product. We have also added a reference to the EXIM documentation. Indeed, EXIM uses the kinematic extrapolation method explained in the text below.

   L. 284 "In addition to U-Net, we consider in our comparison, a product  called EXIM (for Extrapolated Imagery), developed by EUMETSAT as part of their NWCSAF/GEO products [García-Pereda et al., 2019]. This product involves applying the Atmospheric Motion Vector field multiple times to a current image to produce forecasts. Each pixel's new location is calculated using the motion vector, and this process is repeated, assuming a constant displacement field. For continuous variables like brightness temperature, the method uses weighted contributions to forecast pixel values, ensuring that there are no gaps by interpolating values from adjacent pixels if necessary. For categorical variables such as cloud type, the pixel value is directly assigned to the new location, and conflicts are resolved by overwriting. If

a pixel is not touched by any trajectory, the value is determined by the majority class of its nearest neighbours [García-Pereda et al., 2019][1]. This approach is also called kinematic extrapolation."

L. 325 "This observation aligns with the fact that the Persistence uses the last observation as its predictions, and EXIM is advecting the last observation  using the kinematic extrapolation, which keeps the same level of details without diffusion effects [García-Pereda et al., 2019]."

Kind regards,
The authors.

**References**

[García-Pereda et al., 2019] García-Pereda, J., Fernandez-Serdan, J. M., Alonso, O., Sanz, A., Guerra, R., Ariza, C., Santos, I., and Fernández, L. (2019). NWCSAF High Resolution Winds (NWC/GEO-HRW) Stand-Alone Software for Calculation of Atmospheric Motion Vectors and Trajectories. *Remote Sensing*, 11(17):2032. Number: 17 Publisher: Multidisciplinary Digital Publishing Institute.
* * *
[1]https://www.nwcsaf.org/exim_description (last visit 4 July 2024)